# Distinct SoxB1 networks are required for naïve and primed pluripotency

Andrea Corsinotti[1,2†], Frederick CK Wong[1†‡], Tülin Tatar[1], Iwona Szczerbinska[1§#], Florian Halbritter[1¶], Douglas Colby[1], Sabine Gogolok[1], Raphaël Pantier[1], Kirsten Liggat[1], Elham S Mirfazeli[1], Elisa Hall-Ponsele[1], Nicholas P Mullin[1], Valerie Wilson[1]*, Ian Chambers[1]*

[1]MRC Centre for Regenerative Medicine, Institute for Stem Cell Research, School of Biological Sciences, University of Edinburgh, Edinburgh, Scotland; [2]Department of Anatomy and Embryology, Faculty of Medicine, University of Tsukuba, Ibaraki, Japan

*For correspondence:
v.wilson@ed.ac.uk (VW);
ichambers@ed.ac.uk (IC)

[†]These authors contributed equally to this work

Present address: [‡]The WT/CRUK Gurdon Institute, University of Cambridge, Cambridge, England; [§]Gene Regulation Laboratory, Genome Institute of Singapore, Singapore, Singapore; [#]Department of Biochemistry, National University of Singapore, Singapore, Singapore; [¶]CeMM Research Center for Molecular Medicine of the Austrian Academy of Sciences, Vienna, Austria

Competing interests: The authors declare that no competing interests exist.

**Abstract** Deletion of *Sox2* from mouse embryonic stem cells (ESCs) causes trophectodermal differentiation. While this can be prevented by enforced expression of the related SOXB1 proteins, SOX1 or SOX3, the roles of SOXB1 proteins in epiblast stem cell (EpiSC) pluripotency are unknown. Here, we show that *Sox2* can be deleted from EpiSCs with impunity. This is due to a shift in the balance of SoxB1 expression in EpiSCs, which have decreased Sox2 and increased Sox3 compared to ESCs. Consistent with functional redundancy, *Sox3* can also be deleted from EpiSCs without eliminating self-renewal. However, deletion of both *Sox2* and *Sox3* prevents self-renewal. The overall SOXB1 levels in ESCs affect differentiation choices: neural differentiation of *Sox2* heterozygous ESCs is compromised, while increased SOXB1 levels divert the ESC to EpiSC transition towards neural differentiation. Therefore, optimal SOXB1 levels are critical for each pluripotent state and for cell fate decisions during exit from naïve pluripotency.
DOI: https://doi.org/10.7554/eLife.27746.001

## Introduction

Pluripotent cells have the unique ability to differentiate into every cell type of an adult organism (*Nichols and Smith, 2009*). During mouse development, the embryo contains pluripotent cells in the epiblast until the onset of somitogenesis (*Osorno et al., 2012*; *Chambers and Tomlinson, 2009*). Distinct pluripotent cell types can be isolated in culture from the preimplantation and postimplantation epiblast. Embryonic stem cells (ESCs) (*Evans and Kaufman, 1981*; *Martin, 1981*) from preimplantation embryos are termed 'naïve' pluripotent cells. Epiblast stem cells (EpiSCs) (*Tesar et al., 2007*; *Brons et al., 2007*), commonly isolated from the post-implantation epiblast, are known as 'primed' pluripotent cells. Naïve and primed cells differ dramatically in responses to extracellular signals (*Nichols and Smith, 2009*). ESCs self-renew in response to a combination of leukaemia inhibitory factor (LIF) and either foetal calf serum (FCS), bone morphogenic protein (BMP) or Wnt (*Smith et al., 1988*; *Ying et al., 2003a*; *ten Berge et al., 2011*) and differentiate in response to fibroblast growth factor (FGF) (*Kunath et al., 2007*; *Stavridis et al., 2007*). In contrast, FGF together with Nodal/Activin A are required for EpiSCs self-renewal (*Vallier et al., 2009*; *Guo et al., 2009*).

Pluripotency is regulated by a pluripotency gene regulatory network (PGRN) (*Chambers and Tomlinson, 2009*; *Festuccia et al., 2013*; *Wong et al., 2016*). While some transcription factors such as Esrrb and T (Brachyury) are associated more closely with naïve or primed pluripotent states respectively (*Festuccia et al., 2012*; *Osorno et al., 2012*; *Tsakiridis et al., 2014*), the core TFs of the PGRN, Nanog, Sox2 and Oct4 (the *Pou5f1* gene product, also referred as Oct3/4) are expressed

in both naïve and primed pluripotent cells (*Niwa et al., 2000*; *Masui et al., 2007*; *Avilion et al., 2003*; *Chambers et al., 2003*; *Karwacki-Neisius et al., 2013*; *Osorno et al., 2012*; *Festuccia et al., 2012*; *Brons et al., 2007*; *Tesar et al., 2007*). While the role of Sox2 has been extensively characterised in naïve cells (*Wong et al., 2016*), its role in primed pluripotency is less well known.

Sox2 is a member of a family of twenty Sox TFs (*Pevny and Lovell-Badge, 1997*; *Kamachi and Kondoh, 2013*). All SOX proteins contain a High-Mobility-Group (HMG) box DNA-binding domain closely related to the founding member of the Sox family, SRY (*Kondoh and Lovell-Badge, 2016*). While some SOX proteins contain a transcriptional activation domain, others contain repression domains (*Uchikawa et al., 1999*; *Bowles et al., 2000*; *Ambrosetti et al., 2000*). The paradigm of action for SOX proteins is that they bind to target gene sequences through a DNA-mediated interaction with a partner protein, to specify target gene selection (*Kamachi et al., 1999*; *Reményi et al., 2003*; *Williams et al., 2004*; *Kamachi and Kondoh, 2013*). In pluripotent cells the principal interaction of SOX2 with OCT4 (*Ambrosetti et al., 1997*, *2000*) is considered to positively regulate expression of many pluripotency-specific genes including *Nanog*, *Oct4* and *Sox2* (*Tomioka et al., 2002*; *Chew et al., 2005*; *Okumura-Nakanishi et al., 2005*; *Rodda et al., 2005*; *Kuroda et al., 2005*). Loss of SOX2 in ESCs induces trophoblast differentiation, phenocopying OCT4 loss and supporting the idea of a mutually dependent mode of action (*Niwa et al., 2000*; *Masui et al., 2007*).

Analysis of sequence conservation within the HMG box has divided the Sox family into eight groups that can be further divided into subgroups based on homology outside the HMG box (*Kondoh and Lovell-Badge, 2015*; *Kamachi, 2016*). SOX1, SOX2 and SOX3 belong to the SOXB1 group and also contain transcriptional activation domains (*Uchikawa et al., 1999*; *Ambrosetti et al., 2000*; *Bowles et al., 2000*; *Kondoh and Kamachi, 2010*; *Ng et al., 2012*; *Kamachi and Kondoh, 2013*). SOXB1 proteins bind the same DNA sequence in vitro (*Kamachi et al., 1999*; *Kamachi, 2016*). Previous studies demonstrated that SOXB1 factors are co-expressed during embryonic development and can substitute for each other in different biological systems, both in vitro and in vivo (*Wood and Episkopou, 1999*; *Niwa et al., 2016*; *Adikusuma et al., 2017*). Here, we investigate the requirements of naïve and primed pluripotent states for SOXB1 expression. Our results indicate that the essential requirement of SOXB1 function for naïve pluripotent cells extends to primed pluripotent cells. SOX3, which is highly expressed in primed pluripotent cells, functions redundantly with SOX2, rendering SOX2 dispensable in these cells. We further provide evidence that critical SOXB1 levels are required to specify the identity of cells exiting the naïve pluripotent state.

## Results

### A fluorescent reporter of SOX2 protein expression

To investigate the expression of Sox2 in pluripotent cells, a live cell reporter that retained Sox2 function was prepared by replacing the *Sox2* stop codon with a T2A-H2B-tdTomato cassette (*Figure 1A*; *Figure 1—figure supplement 1A*). Correctly targeted cells were identified by Southern analysis and are referred to as E14Tg2a-Sox2-tdTomato (TST) cells (*Figure 1—figure supplement 1B*). Fluorescence microscopy of targeted cells showed a close correlation between SOX2 and tdTomato levels (*Figure 1—figure supplement 2*). Moreover, tdTomato expression recapitulated the SOX2 expression pattern in chimeric embryos (*Figure 1—figure supplement 3*). Targeted cells also showed the expected morphological differences when cultured in a combination of LIF plus inhibitors of MEK and GSK3β (LIF/2i), in LIF/FCS, in LIF/BMP or after passaging in Activin/FGF (*Figure 1A*). These results indicate that TST cells behave normally and provide a useful live cell report of Sox2 expression levels.

TST ESCs were next assessed by fluorescence microscopy and FACS. In LIF/2i, tdTomato expression was high and unimodal (*Figure 1A*). In LIF/FCS or LIF/BMP the same predominant high-expressing population was present but with a shoulder of reduced expression and a small number of tdTomato-negative cells, which appeared to coincide with morphologically differentiated cells (*Figure 1A*). In continuous culture in Activin/FGF, tdTomato expression was bimodal with the highest expression levels overlapping the lower expression levels seen in ESCs cultured in LIF/FCS or LIF/BMP (*Figure 1A,B*).

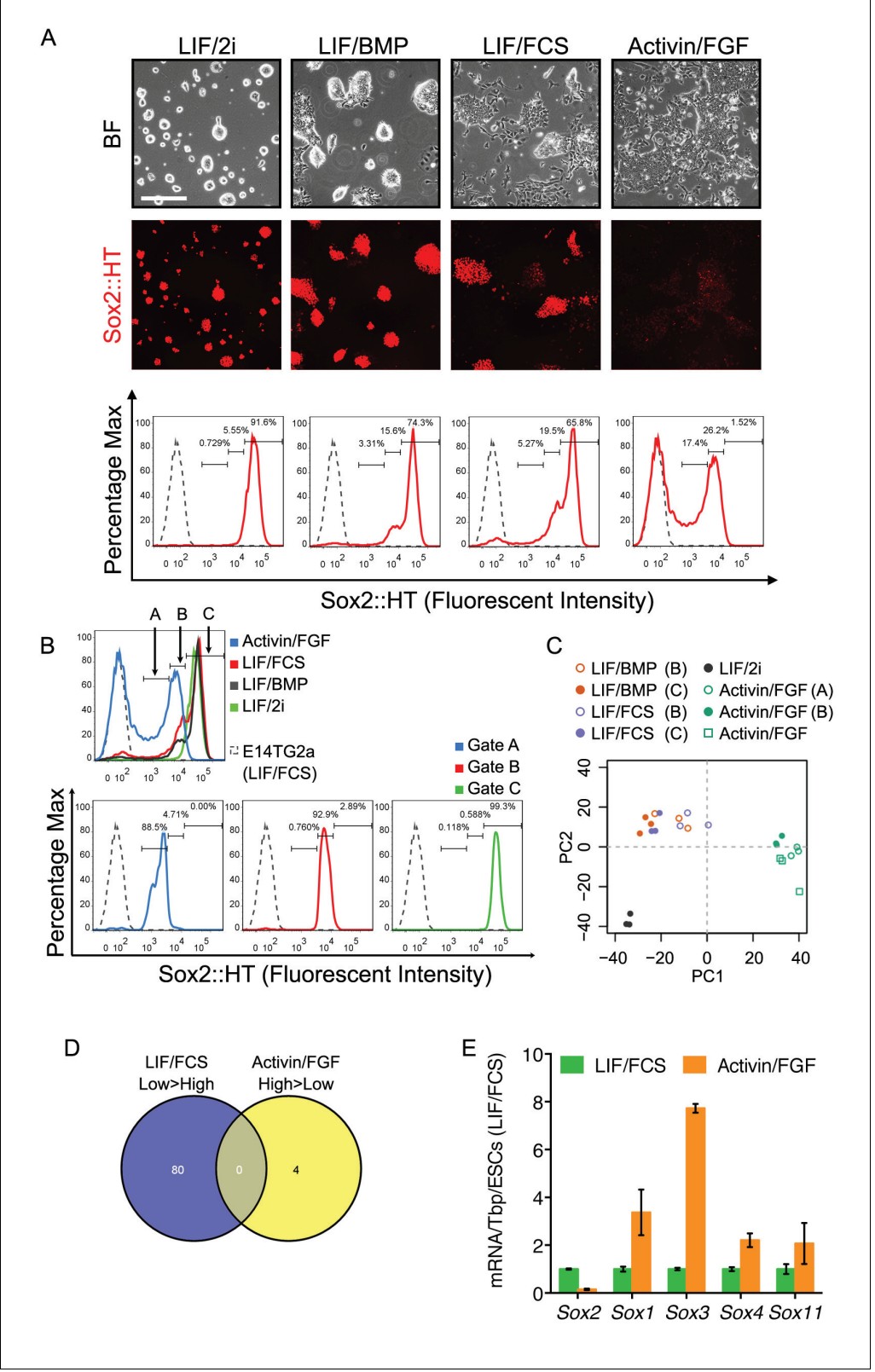

**Figure 1.** Different roles of Sox2 in preimplantation and postimplantation pluripotency. (**A**) Expression of the Sox2-T2A-H2b-tdTomato (Sox2::HT) reporter from the endogenous *Sox2* allele in targeted TST18 cells. TST18 cells cultured in LIF/FCS/GMEMβ were replated in LIF/2i/N2B27 or LIF/BMP4/N2B27 for four passages or in Activin/FGF/N2B27 (Activin/FGF) for nine passages, examined microscopically (top) and assessed by flow cytometry

*Figure 1 continued on next page*

*Figure 1 continued*

(bottom); E14Tg2a cells were represented as a grey dashed line. (B) Three gates (A, B, C) were used to purify cells for microarray analysis. Gate C captured the Sox2::HT level in LIF/2i cultured ESCs. Gate B captured the overlapping Sox2::HT level in LIF/FCS, LIF/BMP and Activin/FGF cultured cells. Gate A captured the lower Sox2::HT level in Activin/FGF. (C) Principal component analysis of cells in different culture conditions, either unsorted or sorted using the gates indicated by brackets. (D) Despite similar Sox2::HT levels, no differentially expressed genes (DEGs, FDR = 0.1) were common to LIF/FCS-low and Activin/FGF-high cell populations. (E) RT-qPCR analysis of the indicated transcripts in ESCs (LIF/FCS, red bars) and EpiSCs (Activin/FGF, cultured for 16 passages, blue bars). Transcript levels were normalized to *Tbp* and plotted relative to ESCs. Error bars represent standard error of the mean (n = 3 to 4).

DOI: https://doi.org/10.7554/eLife.27746.002

The following figure supplements are available for figure 1:

**Figure supplement 1.** (A) Targeting T2A-H2b-tdTomato to the *Sox2* locus stop codon yields the Sox2::HT fluorescent reporter allele.
DOI: https://doi.org/10.7554/eLife.27746.003

**Figure supplement 2.** Imaging of E14Tg2a and TST18 ESCs for SOX2 by immunofluorescence (green) and of Sox2::HT fluorescence (red); DAPI is grey.
DOI: https://doi.org/10.7554/eLife.27746.004

**Figure supplement 3.** TST18 cells transfected with a CAG-GFP constitutive reporter were aggregated with isolated morulae and chimeric embryos assessed at the indicated stages for contribution of TST18 cells.
DOI: https://doi.org/10.7554/eLife.27746.005

**Figure supplement 4.** Differentially expressed genes common to LIF/FCS-low and Activin/FGF-low were not associated with any gene ontology (GO) terms using high-stringency GO term clustering analysis.
DOI: https://doi.org/10.7554/eLife.27746.006

**Figure supplement 5.** Common differentially expressed genes (DEGs, FDR = 0.1) enriched in LIF/FCS-high, LIF/BMP-high and DEGs positively correlated to Sox2 (r2 $\geq$0.9).
DOI: https://doi.org/10.7554/eLife.27746.007

**Figure supplement 6.** Microarray signal intensity of *Sox* family members in all sorted populations.
DOI: https://doi.org/10.7554/eLife.27746.008

## Gene expression in ESCs and EpiSCs expressing distinct Sox2 levels

ESCs and EpiSCs expressing distinct Sox2 levels were separated by FACS according to the tdTomato level (*Figure 1B*). Microarray analysis was then used to compare gene expression in cells from different culture conditions expressing similar Sox2 levels and in cells from the same cultures expressing distinct Sox2 levels (*Supplementary file 1*). Cells expressing the highest tdTomato levels in LIF/2i, LIF/FCS or LIF/BMP cultures were purified using gate C. The mid-level expression gate B enabled purification of the Sox2-low fraction of ESCs cultured in LIF/FCS or LIF/BMP as well as the highest Sox2-expressing EpiSCs. EpiSCs were also purified using the lower expression gate A. Re-sorting confirmed effective purification prior to RNA extraction and analysis (*Figure 1B*). Compared to the Sox2-high population, Sox2-low EpiSCs had upregulated differentiation markers. Transcripts expressed by Sox2-low EpiSCs differed from those expressed by Sox2-low ESCs (*Figure 1—figure supplement 4*; *Supplementary file 1*) suggesting that ESCs and EpiSCs have distinct differentiation propensities. ESCs from both LIF/BMP and LIF/FCS cultures sorted for highest tdTomato expression (gate C) were enriched for naïve pluripotency transcripts with mRNAs common to both (including *Nr5a2*, *Tbx3* and *Tcl1*) positively correlating to Sox2 in all samples (*Figure 1—figure supplement 5*). Moreover, principal component analysis indicated that these Sox2-high ESCs cluster together and separately from ESCs cultured in LIF/2i (*Figure 1C*) as seen by others (*Boroviak et al., 2015*). More importantly, principal component analysis also showed that EpiSCs and ESCs expressing the same Sox2 level (gate B) were transcriptionally distinct (*Figure 1C*) and shared no commonly enriched mRNAs (*Figure 1D*). Thus, Sox2 levels alone do not dictate the distinction between ESC and EpiSC states.

Microarray and quantitative transcript analyses highlighted changes in Sox gene expression levels (*Figure 1E*, *Figure 1—figure supplement 6*). Although other Sox gene expression changes occurred, the ability of SOXB1 proteins to function redundantly in ESC self-renewal (*Niwa et al., 2016*) prompted us to assess the capacity of SOXB1-related proteins to function more widely in pluripotent cells.

## A subset of SOX family proteins can functionally replace *Sox2* in ESC self-renewal

Recently, *Niwa et al. (2016)* reported that ESC self-renewal could be maintained in the absence of SOX2 by expression of other SOXB1 and SOXG proteins (*Niwa et al., 2016*). We addressed the question of functional redundancy with Sox2 in ESCs using an independent *Sox2* conditional knock-out (SCKO) ESC line (*Favaro et al., 2009*; *Gagliardi et al., 2013*). SCKO ESCs have one *Sox2* allele replaced by β-geo, the second allele flanked by *loxP* sites and constitutively express a tamoxifen-inducible Cre recombinase (CreER$^{T2}$) (*Figure 2A*). SCKO ESCs were transfected with a plasmid in which constitutive Sox cDNA expression was linked to hygromycin B resistance and were either left untreated or were simultaneously treated with tamoxifen to excise *Sox2* (*Figure 2A*). ESC self-renewal was assessed after 8–10 days of hygromycin selection. Transfection of plasmids encoding transcriptional activator proteins of the SOXB1 family (SOX1, SOX2 or SOX3) enabled similar levels of ESC colony formation in the absence of *Sox2* (*Figure 2B,C*). In contrast, the SOXB2 proteins (SOX14 and SOX21), in which the SOXB DNA-binding domain is linked to transcriptional repression domains, did not direct undifferentiated ESC colony formation (*Figure 2C*). Of the other SOX proteins tested, only SOX15 showed any capacity to direct undifferentiated ESC colony formation (*Figure 2C*), as previously described (*Niwa et al., 2016*), possibly because it has the most similar DNA-binding domain to SOXB proteins (*Kamachi and Kondoh, 2013*). These results indicate that ESC self-renewal requires the function of a SOX protein with a SOXB-like DNA-binding domain coupled to a transcriptional activating function.

To further assess the ability of SOX proteins to sustain ESC self-renewal, we attempted to expand transfected cell populations. PCR analysis confirmed deletion of the *Sox2* conditional allele (400 bp) from expanded cell populations expressing SOX1, SOX2 or SOX3 (*Figure 2—figure supplement 1*). Furthermore, SoxB1-rescued (S1R and S3R) SKO ESCs retained OCT4 and NANOG expression, similar to SCKO and Sox2-rescued SKO (S2R) ESCs (*Figure 2—figure supplement 2*). The transgene expression levels in ESC populations rescued by Sox1, 2 or 3 were assessed by quantitative transcript analysis using common primers across the CAG intron. This demonstrated that the Sox3 transgene mRNA was expressed at a higher level than the Sox1 or Sox2 transgenes (*Figure 2D*). These results confirm the findings of Niwa *et al.* in an independent cell line (*Niwa et al., 2016*).

## *Sox3* is dispensable for both naïve and primed pluripotency

*Sox3* transcripts are present in ESCs at lower levels than Sox2 (*Figure 3—figure supplement 1*) and a previous report has suggested that Sox3 is dispensable for ESC self-renewal (*Rizzoti et al., 2004*; *Rizzoti and Lovell-Badge, 2007*). To directly assess this possibility, and as a first step to determining whether Sox3 is required in EpiSCs, CRISPR/Cas9 was used to delete the *Sox3* gene from male E14Tg2a ESCs (*Hooper et al., 1987*; *Doetschman et al., 1987*) using two small guide RNAs (sgRNA1 and sgRNA2) (*Figure 3A*). After targeting, E14Tg2a ESCs were plated at low density and single clones were isolated. PCR genotyping using primers flanking the predicted Cas9 cut sites (*Figure 3A*) identified two clones (S3KO8 and S3KO35) in which *Sox3* had been deleted (*Figure 3B*). Replating at clonal density indicated that both clones retained an efficient self-renewal ability (*Figure 3C*). Moreover, both clones had unchanged levels of *Nanog*, Oct4 and *Sox2* mRNAs (*Figure 3D*). These findings indicate that *Sox3* is dispensable for ESC self-renewal, confirming previously unpublished data (*Rizzoti et al., 2004*).

Next, the requirement of SOX3 for primed pluripotency was determined by examining the ability of the above *Sox3* knockout ESC clones, together with parental E14Tg2a ESCs, to be converted to EpiSCs by serial passaging in Activin and FGF (*Guo et al., 2009*). Quantitative transcript analysis at passage 12 indicated similar levels of *Sox2*, Oct4 and *Nanog* mRNA expression in wild-type and *Sox3* knockout EpiSCs (*Figure 3E*). In contrast, *Sox1* mRNA levels were reduced in both *Sox3* knockout clones (*Figure 3E*) and T (Brachyury) was more variably expressed (*Figure 3E*). The ability of EpiSCs to self-renew in the absence of SOX3 was maintained over 25 passages without affecting SOX2 and OCT4 protein expression or EpiSC morphology (*Figure 3F,G*). These data demonstrate that both naïve and primed pluripotent cells can self-renew in the absence of Sox3.

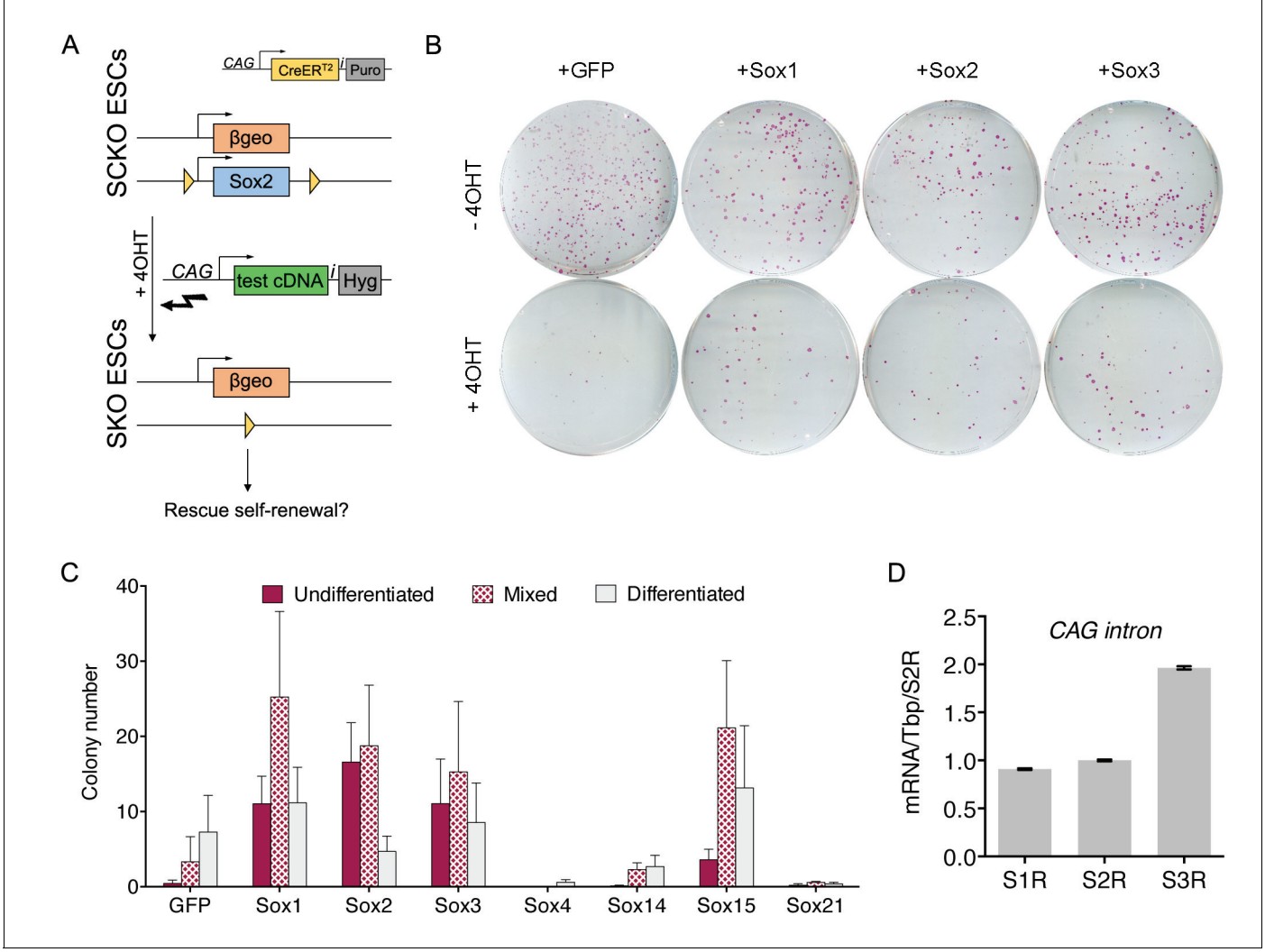

**Figure 2.** The ESC function of Sox2 can be substituted by SOXB1 and SOXG proteins. (**A**) Strategy for testing the ability of candidate Sox cDNAs to rescue ESC self-renewal upon *Sox2* deletion. *Sox2*^fl/-^ (SCKO) ESCs were treated with 4-hydroxy-tamoxifen (4OHT) to induce nuclear localisation of CreER^T2^ and consequent *loxP*-mediated *Sox2* excision. Simultaneous transfection of test cDNAs linked to hygromycin phosphotransferase via an IRES (i) were tested for Sox2 complementation activity. (**B**) SCKO ESCs transfected with the indicated cDNAs were cultured in the presence of hygromycin B and in the absence (−4OHT) or presence (+4 OHT) of 4-hydroxyl-tamoxifen for 8 days before being fixed and stained for alkaline phosphatase (AP) activity. (**C**) Stained colonies were scored based on AP+ (undifferentiated), mixed or AP− (differentiated) morphology. Error bars represent standard error mean (n = 3). (**D**) RT-qPCR analysis of the rescuing transgenes in Sox1-, Sox2- and Sox3-rescued (S1R, S2R, S3R) ESC populations grown in the presence of 4-hydroxyl-tamoxifen as described in *Figure 2A,B*. RT-qPCR was performed using common primers designed across the CAG intron upstream of the rescuing transgene. Values were normalised over *Tbp* and expressed relative to S2R cells. Error bars represent the standard error of the mean (n = 3).

DOI: https://doi.org/10.7554/eLife.27746.009

The following figure supplements are available for figure 2:

**Figure supplement 1.** (TOP) Strategy for genotyping the *Sox2* deletion by PCR.

DOI: https://doi.org/10.7554/eLife.27746.010

**Figure supplement 2.** Immunofluorescence staining of SOX2, NANOG and OCT4 in SCKO, S1R, S2R and S3R ESCs.

DOI: https://doi.org/10.7554/eLife.27746.011

## *Sox2* is dispensable for the maintenance of primed pluripotency

To investigate whether primed pluripotency can be maintained in the absence of SOX2, SCKO EpiSCs were derived by in vitro culture of SCKO ESCs (*Guo et al., 2009*). SCKO EpiSCs were then transfected with a plasmid encoding both Cre recombinase and tdTomato (*Figure 4A*), since the

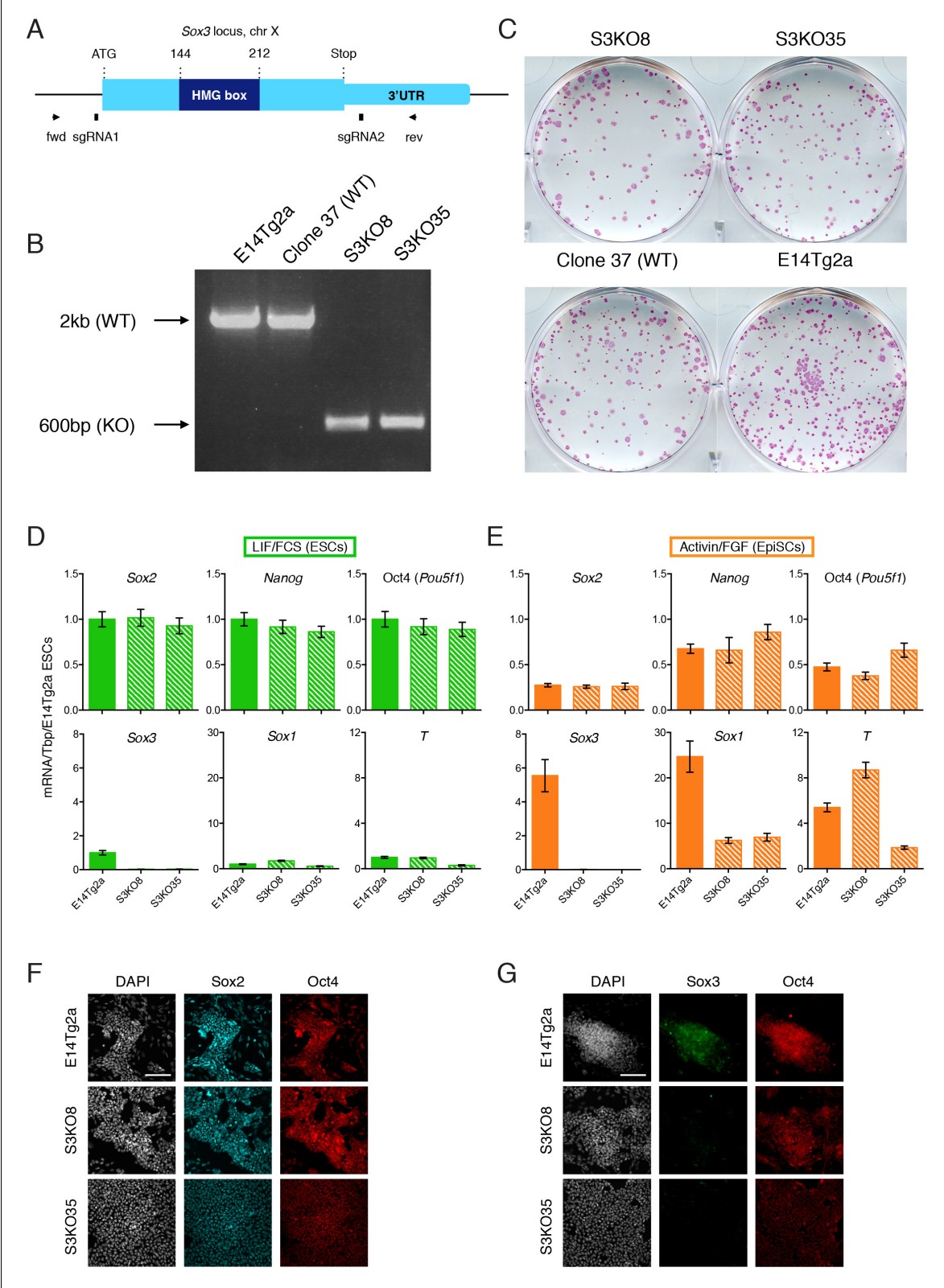

**Figure 3.** *Sox3* is dispensable for pluripotent cell maintenance. (**A**) Schematic representation of the *Sox3* locus on the mouse X chromosome showing the CRISPR/Cas9 strategy for the generation and genotyping of *Sox3* knockout ESCs. The positions of sgRNA1 and sgRNA2 used for the deletion of the *Sox3* locus are indicated alongside the positions of PCR primers used for genotyping. (**B**) PCR genotyping of parental ESCs (E14Tg2a), a wild-type clone (37) and two *Sox3* knockout clones (S3KO8, S3KO35) after CRISPR/Cas9 targeting of the *Sox3* locus. Band sizes for the WT (~2 kb) and targeted

*Figure 3 continued on next page*

*Figure 3 continued*

(~600 bp) alleles are shown. (C) Alkaline phosphatase staining in S3KO8, S3KO35 and WT (Clone 37 and E14Tg2a) ESCs grown at clonal density for 7 days. (D–E) RT-qPCR analysis of the indicated transcripts in wild-type E14Tg2a and *Sox3*−/Y (S3KO8, S3KO35) ESCs (D) and EpiSCs (E) derived by serial passaging (>10 passages) in Activin/FGF conditions. Error bars represent standard error of the mean (n = 3). (F-G) Immunofluorescence staining of SOX2 (F), SOX3 (G) and OCT4 proteins in E14Tg2a and *Sox3*−/Y EpiSCs (S3KO8, S3KO35). Scale bar, 100 μm.

DOI: https://doi.org/10.7554/eLife.27746.012

The following figure supplement is available for figure 3:

**Figure supplement 1.** RT-qPCR analysis of the indicated transcripts in E14Tg2a and SCKO ESCs.

DOI: https://doi.org/10.7554/eLife.27746.013

CAG CreERT$^2$ transgene (*Figure 2A*) had been silenced during the EpiSC transition. After 12–24 hr, tdTomato-positive EpiSCs were sorted by FACS and re-plated at low cell density to allow expansion of single clones. PCR genotyping of expanded EpiSC clones revealed that the *loxP*-flanked *Sox2* allele had been excised generating *Sox2*−/− (SKO) clones (*Figure 4A*). Immunoblot analysis confirmed that these SKO clones did not express SOX2 protein and that *Sox2*fl/- EpiSCs expressed SOX2 at 50% the level of *Sox2*+/+ EpiSCs (*Figure 4B*). Moreover, NANOG protein levels were similar to parental SCKO EpiSCs or wild-type E14Tg2a EpiSCs (*Figure 4B*). *Sox2*−/− EpiSCs retained an undifferentiated morphology (*Figure 4C*) and immunostaining confirmed the absence of SOX2 and continued OCT4 expression compared to parental SCKO EpiSCs (*Figure 4D*). Quantitative transcript analysis of Oct4, *Nanog* and *Sox2* , indicated that, while *Sox2* mRNA was absent from *Sox2*−/− EpiSCs, both Oct4 and *Nanog* mRNA levels were unaffected (*Figure 4E*). This suggests that the core pluripotency gene regulatory network remains active even in the absence of SOX2 and that, in contrast to the situation with naïve pluripotency, primed pluripotency is not critically dependent upon Sox2.

The ability of SOXB1 proteins to substitute for one another in ESCs raised the hypothesis that functional redundancy between SOX2 and SOX3 may be responsible for the maintenance of pluripotency in primed EpiSCs. Expression of both *Sox1* and *Sox3* mRNAs was increased in EpiSCs compared to ESCs (*Figure 1E*). Examination of *Sox2*−/− EpiSCs showed that while Sox1 mRNA expression was reduced, Sox3 mRNA levels were elevated compared to control cells (*Figure 4E*). This suggests that Sox3 expression might functionally compensate for the lack of Sox2 in *Sox2*−/− EpiSCs, as is the case in *Sox2*-null E14.5 forebrain, where a 2-fold increase in *Sox3* mRNA was detected (*Miyagi et al., 2008*). Alternatively, SOXB1 proteins may be irrelevant for EpiSC self-renewal. To distinguish between these possibilities we developed an approach to gene disruption using CRISPR/Cas9.

## Testing the functional effects of Sox ORF disruption using CRISPR/Cas9

To examine the functional dependence of cells on Sox genes, a CRISPR/Cas9 approach was developed and tested initially using *Sox2*. Insertions or deletions (indels) into the *Sox2* open reading frame (ORF) were introduced immediately upstream of the sequence encoding the SOX2 HMG domain (*Figure 5—figure supplement 1A*). Out-of-frame indels in this position are expected to abolish the DNA-binding ability of any aberrant protein that might still be produced. The frequency and length of indels induced by CRISPR/Cas9-mediated targeting of the endogenous *Sox2* locus were investigated by TIDE (Tracking of Indels by Decomposition) analysis (*Brinkman et al., 2014*). *Sox2*fl/- SCKO ESCs that constitutively express Sox1, Sox3 or GFP (generated in *Figure 2B* in the absence of tamoxifen) were analysed. If, as expected, Sox1 or Sox3 can rescue self-renewal, then such cells would carry both in-frame and out-of-frame (deleterious) indels (*Figure 5—figure supplement 1B*). In contrast, if ESC self-renewal relies on the remaining *Sox2* allele, as anticipated in the case of SCKO ESCs constitutively expressing GFP, then cells carrying deleterious indels should be eliminated from the population. In this case, the only modifications present would be non-deleterious in-frame indels (*Figure 5—figure supplement 1B*). SCKO ESCs expressing Sox1, Sox3 or GFP were transfected with a plasmid encoding the sgRNA and eCas9. Cells were selected, genomic DNA isolated, PCR amplified, sequenced and analysed by TIDE. The population of SCKO ESCs expressing GFP contained *Sox2* loci with no out-of-frame indels detected (*Figure 5—figure supplement 2*). In contrast, SCKO ESCs constitutively expressing Sox1 or Sox3 tolerated out-of-frame, deleterious indels at *Sox2* (size +1,−10, −13) (*Figure 5—figure supplement 2*). These results show that indel

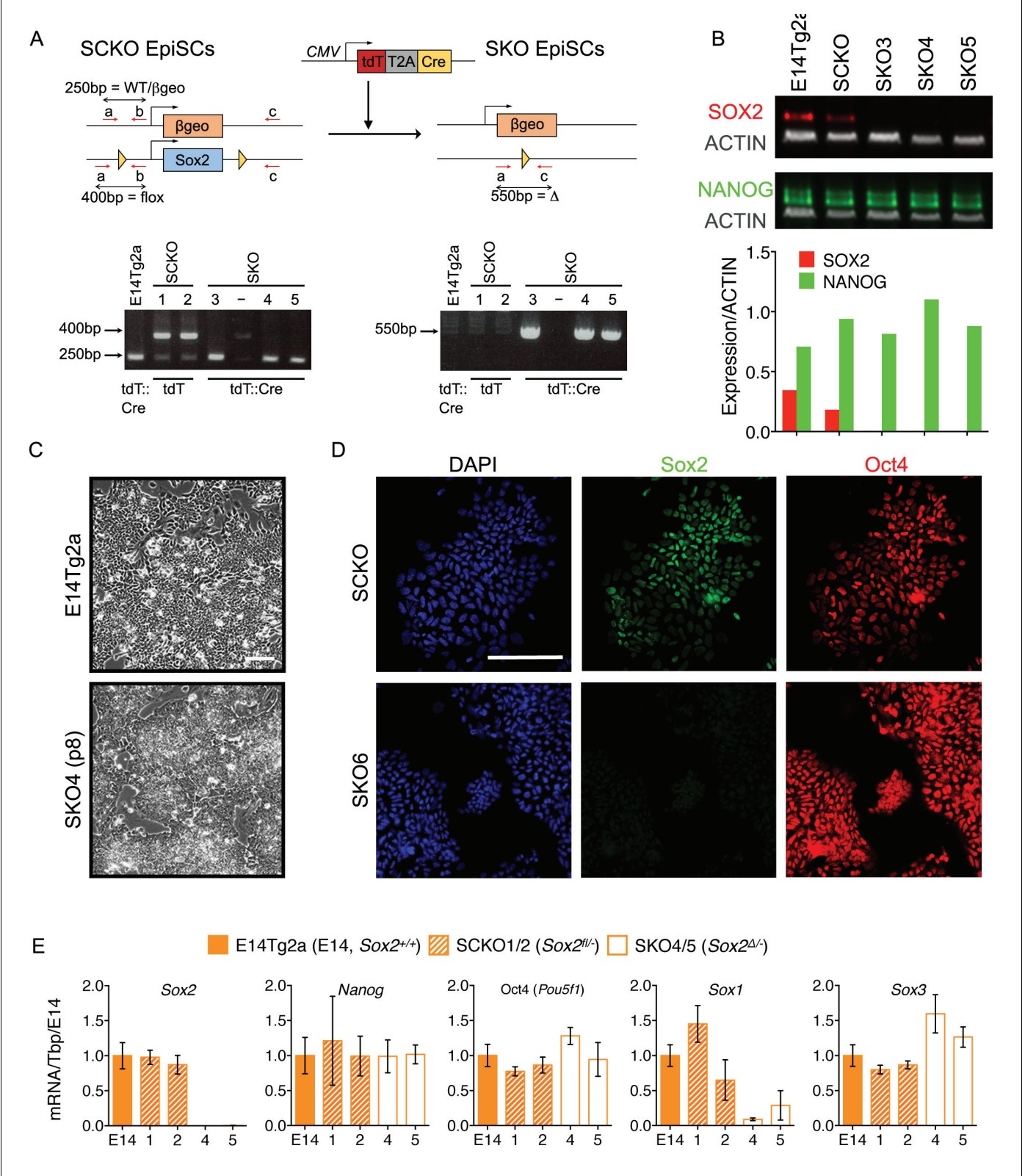

**Figure 4.** *Sox2* is dispensable for the maintenance of EpiSCs. (**A**) Strategy for *Sox2* deletion in EpiSCs. *Sox2*^fl/- (SCKO) EpiSCs were transfected with pCMV-tdTomato-T2A-Cre. After 12–24 hours, cells were sorted for tdTomato expression and re-plated in the presence of ROCK inhibitor. Clones were picked and expanded before genotyping by PCR as indicated in *Figure 2—figure supplement 1* . SCKO EpiSCs were also transfected with pCMV-tdTomato (lacking Cre). Expanded clones were numbered as indicated. (**B**) Immunoblot analysis of E14Tg2a, *Sox2*^fl/- (SCKO) and *Sox2*^-/- (SKO) EpiSCs

*Figure 4 continued on next page*

Figure 4 continued

showing that SOX2 protein (red) is reduced in SCKO and absent in SKO EpiSCs. NANOG protein levels (green) were unaffected. Protein levels (normalized to the βactin level) are graphed below. (C) Bright-field morphology of E14Tg2a and SKO4 EpiSCs after 2 weeks (8 passages) in culture. Scale bar, 100 μm. (D) Immunofluorescence staining of SOX2 and OCT4 in SCKO and SKO6 EpiSCs. Scale bar, 100 μm. (E) RT-qPCR analysis of the indicated transcripts in E14Tg2a, Sox2$^{fl/-}$ (SCKO) and Sox2$^{-/-}$ (SKO) EpiSCs. Clone numbers are indicated. Error bars represent the standard error of the mean (n = 2 to 3).
DOI: https://doi.org/10.7554/eLife.27746.014

analysis can be applied to study functional redundancy between SOXB1 group members in the maintenance of pluripotent cells.

## Sox2 and Sox3 are functionally redundant for the maintenance of EpiSCs

Having established the utility of CRISPR/Cas9-mediated indel induction for gene function analysis, a similar strategy was applied to Sox2$^{-/-}$ EpiSCs to determine whether SOXB1 proteins operate in a functionally redundant way to maintain EpiSCs. Since Sox2$^{-/-}$ EpiSCs showed an increase in Sox3 mRNA expression upon deletion of the second Sox2 allele, we focussed on Sox3, which is X-linked and thus present in only one copy in these male EpiSCs (Sox3$^{+/Y}$). Two sgRNAs (sgRNA1 and 2) were designed to independently target the Sox3 ORF immediately upstream of the sequence encoding the HMG DNA-binding domain and tested individually (Figure 5A). TIDE analysis (Figure 5B) showed that in control EpiSCs expressing endogenous Sox2 (E14Tg2a and SCKO) both in frame and out of frame indels within Sox3 were detected using either sgRNA (Figure 5C). In contrast, two independent Sox2$^{-/-}$ EpiSC lines (SKO1, SKO6) retained only in frame deletions that did not disrupt the SOX3 HMG box (Figure 5C). Together with earlier results on Sox2$^{-/-}$ EpiSCs (Figure 4) and Sox3$^{-/Y}$ EpiSCs (Figure 3), these data indicate that EpiSCs lacking either Sox2 or Sox3 alone can be maintained, while cells lacking both SOX2 and SOX3 proteins are lost. Therefore, EpiSC self-renewal requires SOXB1 function, and this can be provided by the expression of either endogenous Sox2 or Sox3.

## Modulating SOXB1 levels affects differentiation and can prevent capture of primed pluripotency

Since Sox2 and Sox3 transcript levels change during ESC to EpiSC differentiation, this raised the question of their importance for attainment of a primed pluripotent state. ESCs that delete Sox2 differentiate to trophectoderm (Masui et al., 2007), while ESCs that continue expressing high SOX2 protein levels during differentiation are biased towards neural fates (Zhao et al., 2004). To assess the effect of increasing the SOXB1 concentration upon differentiation, we examined three clones overexpressing either SOX2 (Figure 6) or SOX3 (Figure 6—figure supplement 1), generated using the approach outlined in Figure 2A. When placed in an EpiSC differentiation protocol, both SOX2- and SOX3-overexpressing clones showed reduced expression of Oct4, Nanog and Nr5a2 transcripts as well as increased expression of Sox1, Mash1 (the Ascl1 gene product) and Pax6 transcripts (Figure 6A; Figure 6—figure supplement 1). Examination of SOX2-overexpressing clones in Activin/FGF showed neural-like cellular morphology (Figure 6B) and βIII-tubulin-positive axonal processes (Figure 6C). These findings indicate that while high SOXB1 levels are tolerated by ESCs, a decreased dosage of SOXB1 is essential for ESCs to transit effectively to an EpiSC state and avoid ectopic neural differentiation.

The SoxB1 requirements during neural differentiation of ESCs were next assessed. Initially, we compared Sox2$^{+/+}$ ESCs previously cultured in LIF/FCS or LIF/2i during neural differentiation induced by culture in N2B27 medium (Ying and Smith, 2003; Ying et al., 2003a). Our results indicate a more rapid induction of a Sox1-GFP reporter (Aubert et al., 2003) and Sox1 mRNA from LIF/FCS cultures than from LIF/2i cultures (Figure 7A). The Sox1-GFP kinetics from 2i/LIF cultures observed here agree with those reported (Marks et al., 2012), although this study noted a slower induction of Sox1-GFP from FCS/LIF cultures. However, as the timing of Sox1-GFP induction from FCS/LIF cultures in previous studies from the same group (Ying et al., 2003a) was consistent with the timings reported here, further differentiation experiments were initiated from LIF/FCS. Expression of

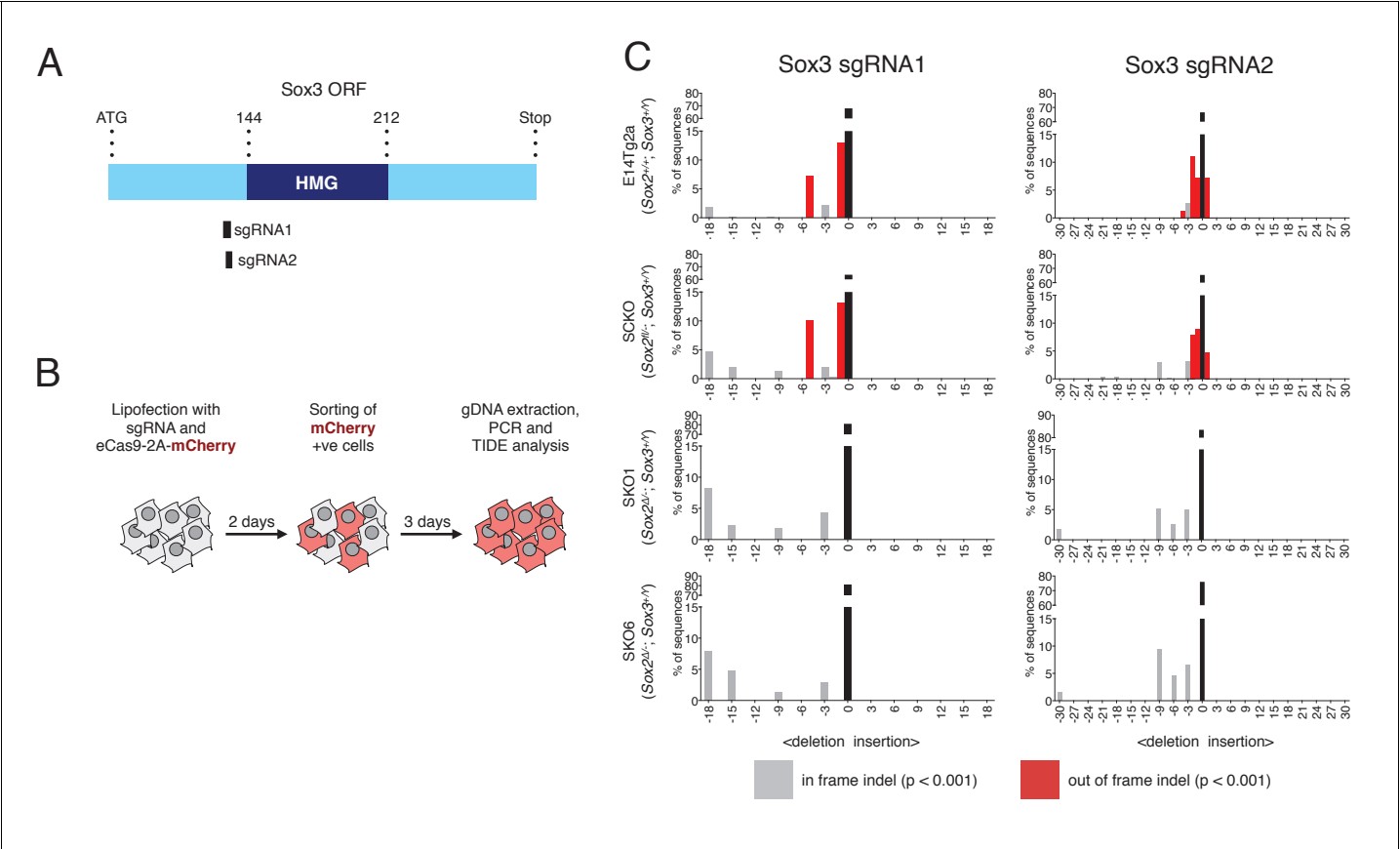

**Figure 5.** Sox3 indel analysis in *Sox2*[-/-] EpiSCs. (A) Schematic representation of the Sox3 ORF with the HMG box highlighted in dark blue. The amino-acid positions of the HMG box relative to the start (ATG) codon of the ORF are shown. The positions of the sgRNAs 1 and 2 used are represented as black bars. (B) Experimental design for Sox3 indel analysis in *Sox2*[-/-] EpiSCs. (C) Indel analysis performed using the TIDE tool (https://tide-calculator.nki.nl/) in two *Sox2*[-/-] EpiSC lines (SKO1 and SKO6), in the parental *Sox2*[fl/-] (SCKO) EpiSCs and in control E14Tg2a EpiSCs using Sox3 sgRNAs 1 and 2. Histograms represent indel frequency and size. Black bars indicate the frequency of unmodified (WT) alleles; grey bars indicate significant in frame indels and red bars indicate significant out of frame indels (p<0.001). Non-significant (n/s, p≥0.001) indels are not shown.

DOI: https://doi.org/10.7554/eLife.27746.015

The following figure supplements are available for figure 5:

**Figure supplement 1.**
DOI: https://doi.org/10.7554/eLife.27746.016

**Figure supplement 2.** Indel analysis performed using the TIDE tool (https://tide-calculator.nki.nl/) in SCKO ESCs expressing either Sox1, Sox3 or GFP transgenes.
DOI: https://doi.org/10.7554/eLife.27746.017

pluripotency markers *Nanog* and *Nr5a2* was decreased in the neural differentiation protocol, although with lower efficiency in *Sox2*[fl/-] ESCs than in control *Sox2*[+/+] cells, while *Fgf5* was induced in both (*Figure 7B*). Strikingly, *Sox1*, Mash1 and *Pax6* mRNAs were induced in *Sox2*[+/+] but not *Sox2*[fl/-] cells (*Figure 7B*), and an increase in *Sox3* was not sustained. These data indicate that a SOXB1 level above that present in *Sox2*[fl/-] cells is required to enable ESCs to undergo neural differentiation in vitro. Furthermore, the expression of *Fgf5* suggests that *Sox2*[fl/-] cells attain some aspects of an early post-implantation identity (*Figure 7B*). This contrasts with the neural differentiation observed in *Sox2*[+/-] mouse embryos (*Avilion et al., 2003*; *Rizzoti and Lovell-Badge, 2007*; *Favaro et al., 2009*), suggesting that compensatory mechanisms other than SOXB1 redundancy exist in the embryo.

To eliminate the possibility that the neural differentiation defect in SCKO cells resulted from a defect unrelated to *Sox2*, CRISPR/Cas9 was used to introduce indels in the *Sox2* ORF in E14Tg2a

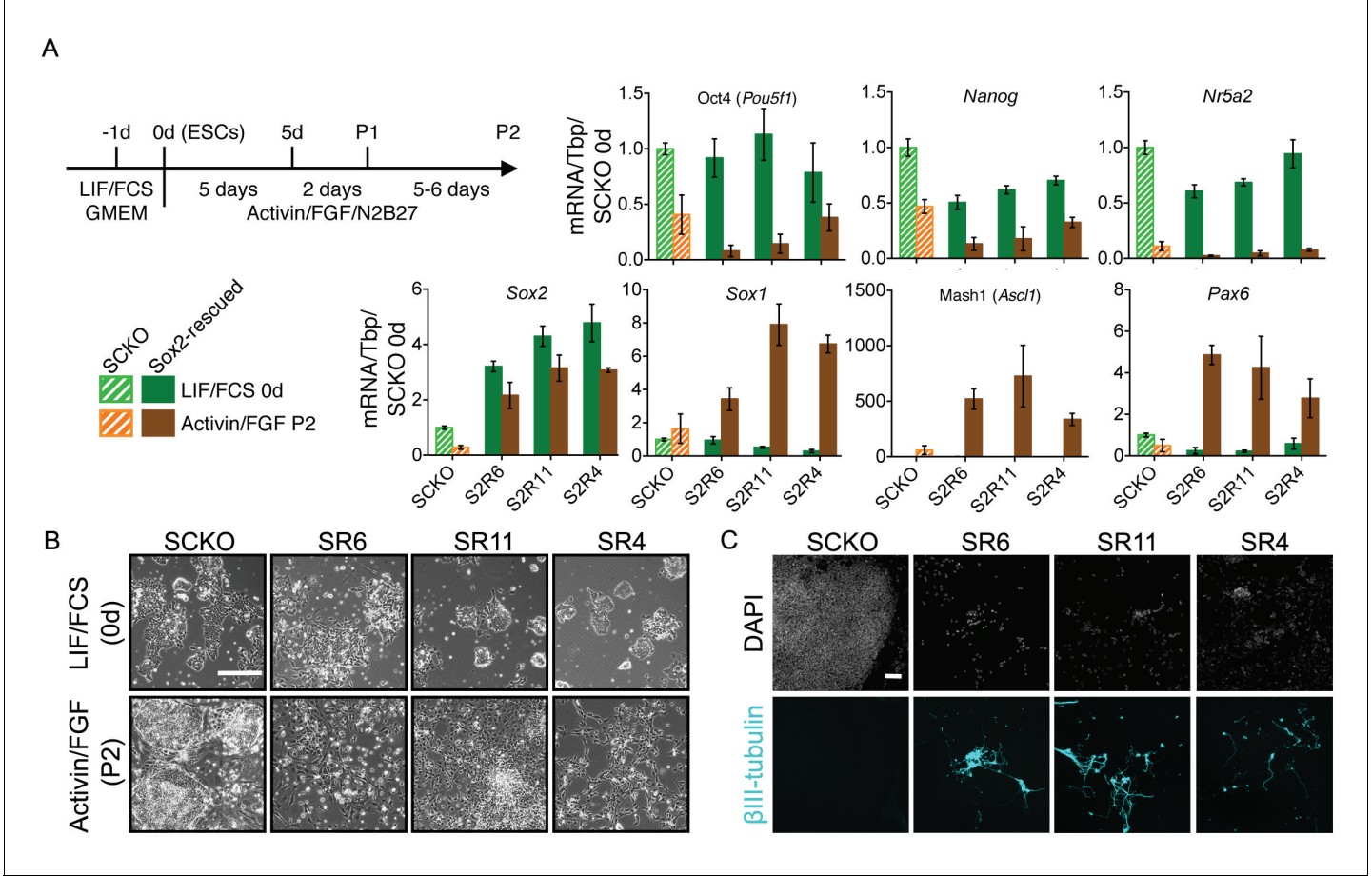

**Figure 6.** Increased SOXB1 levels skew ESC differentiation towards the neural lineage. (**A**) Schematic diagram showing the experimental plan. RT-qPCR analysis of the indicated transcripts in SCKO cells stably transfected and rescued with a Sox2 transgene (12–24 hr prior to a 24 hr treatment with 4OHT) (termed Sox2-rescued, S2R cells). Cells were grown in LIF/FCS conditions or differentiated in Activin/FGF conditions for 13 days (P2). Transcript levels were normalized to *Tbp* and plotted relative to SCKO ESCs. Error bars represent standard error of the mean (n = 3 to 5). (**B**) Bright field images of the indicated cells maintained in LIF/FCS or differentiated in Activin/FGF conditions (P2) . Scale bar, 100 μm. (**C**) Immunofluorescence staining of the neural marker βIII-tubulin (cyan) in cells differentiated in Activin/FGF conditions (P2); DAPI represented in grey. Scale bar, 100 μm.

DOI: https://doi.org/10.7554/eLife.27746.018

The following figure supplement is available for figure 6:

**Figure supplement 1.** RT-qPCR analysis of the indicated transcripts in E14Tg2a cells, *Sox2*[fl/-] cells (SCKO), and *Sox2*[-/-] Sox3-rescued (S3R) cells maintained in LIF/FCS or differentiated in Activin/FGF conditions (P2) .

DOI: https://doi.org/10.7554/eLife.27746.019

ESCs. Using independent sgRNAs, two *Sox2*[+/-] clones were isolated. Clone H2.1 carried an 8 bp deletion on one *Sox2* allele; clone BH1.21 carried a 10 bp deletion on one allele and a 3 bp deletion on the other allele (*Figure 7—figure supplement 1A*). Two additional clones in which the *Sox2* alleles were not modified (H2.14, H2.17) were used as controls. The −8 and −10 deletions cause frame-shifts introducing stop codons, while the −3 bp deletion is likely to be functionally neutral as it occurs N-terminal to the HMG domain (*Figure 7—figure supplement 1B*). Placing *Sox2*[+/+] ESCs (H2.14, H2.17 and E14Tg2a) in a neural differentiation protocol produced βIII-tubulin positive neurons (*Figure 7C*). In contrast, no βIII-tubulin positive cells were detected from parallel treatments of H2.1, BH1.21 and SCKO *Sox2*[+/-] (*Figure 7C*). This establishes that the lack of a functional *Sox2* allele impairs effective neural differentiation of ESCs.

Neural differentiation of pluripotent cells is stimulated by FGF (*Ying et al., 2003a*; *Ying and Smith, 2003*) and inhibited by both BMP and Nodal/Activin (*Ying et al., 2003b*; *Vallier et al., 2004*; *2009*; *Guo et al., 2009*). To determine whether perturbations in these pathways could overcome

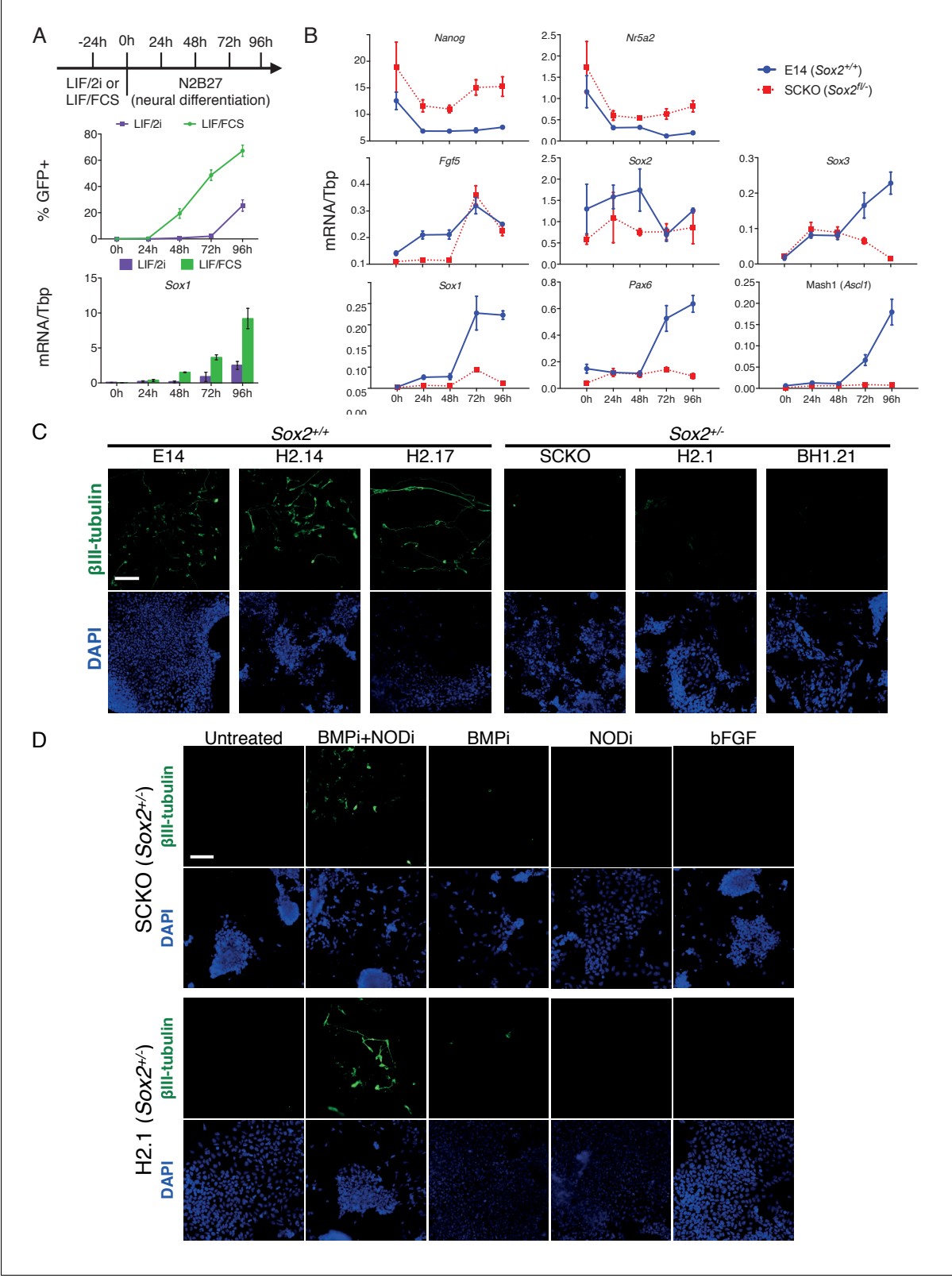

**Figure 7.** Decreased SOX2 levels prevent ESC differentiation into neurons. (**A**) (TOP) Schematic diagram showing experimental plan for neural differentiation. (MIDDLE) Sox1-GFP (*Aubert et al., 2003*) expression in 46C cells cultured in LIF/FCS (green) or LIF/2i (purple) and for the indicated number of hours in N2B27 neural differentiation medium were assessed by flow cytometry. The positive (+) gate was set above the GFP expression level observed in 46C ESCs. The error bars indicate the standard error of the mean (n = 4). (BOTTOM) RT-qPCR analysis of *Sox1* mRNA level in 46C

*Figure 7 continued on next page*

*Figure 7 continued*

cells during neural differentiation. Error bars indicate the standard error of the mean (n = 2). (**B**) RT-qPCR analysis of the indicated transcripts in differentiating E14Tg2a (E14, *Sox2*$^{+/+}$) and SCKO (*Sox2*$^{fl/-}$) cells. Transcript levels were normalized to *Tbp*. Error bars represent standard error of the mean (n = 3). (**C**) Immunofluorescence staining of the neural marker βIII-tubulin (green) in *Sox2*$^{+/+}$ E14Tg2a (E14), H2.14 and H2.17 ESCs, and in *Sox2*$^{+/-}$ SCKO, H2.1 and BH1.21 ESCs differentiated in N2B27 medium for 4 days; DAPI represented in blue. Scale bar, 100 μm. (**D**) Immunofluorescence staining of the neural marker βIII-tubulin (green) in *Sox2*$^{+/-}$and H2.1 ESCs differentiated in N2B27 medium for 4 days in the presence or in the absence of the LDN-193189 BMP inhibitor (BMPi), of the SB-431542 Nodal inhibitor (NODi) and of recombinant bFGF; DAPI represented in blue. Scale bar, 100 μm.

DOI: https://doi.org/10.7554/eLife.27746.020

The following figure supplement is available for figure 7:

**Figure supplement 1.** Genotyping analysis of four ESC clones generated after transfection of CRISPR/Cas9 reagents to induce indels within the *Sox2* ORF in E14Tg2a ESCs.

DOI: https://doi.org/10.7554/eLife.27746.021

the neural differentiation defect of *Sox2*$^{+/-}$ ESCs, H2.1 and SCKO cells were placed in the neural differentiation protocol supplemented with either recombinant bFGF or inhibitors of BMP or Nodal. Additional FGF, or inhibition of Nodal alone, was without effect, while BMP inhibition resulted in only a few βIII-tubulin-positive cells (*Figure 7D*). However, simultaneous BMP and Nodal inhibition enabled *Sox2*$^{+/-}$ ESCs to form βIII-tubulin-positive cells (*Figure 7D*). These results indicate that a reduction in SOX2 levels in ESCs enhances the response of cells to endogenous BMP and Nodal signalling, preventing effective neural differentiation.

To further investigate the effects of modulating SoxB1 gene dosage upon ESC differentiation, *Sox3*$^{-/Y}$ ESCs were generated by CRISPR/Cas9 mediated gene deletion from heterozygous *Sox2*$^{fl/-}$ ESCs (SCKO). PCR genotyping identified two *Sox2*$^{fl/-}$; *Sox3*$^{-/Y}$ ESC clones (#36 and #37) (*Figure 8A*). Quantitative transcript analysis showed that while *Sox2*$^{fl/-}$ ESCs had an expected reduction in *Sox2* mRNA and unchanged levels of *Sox1* and *Sox3* mRNAs compared to E14Tg2a ESCs, deletion of *Sox3* from *Sox2*$^{fl/-}$ ESCs increased *Sox1* and surprisingly, also *Sox2* mRNA to a level similar to that present in E14Tg2a ESCs (*Figure 8B*). These data suggest cross-regulatory interactions between SoxB1 members in which SOX3 protein represses *Sox1* and *Sox2*, either directly or indirectly, in ESCs (*Figure 8B*). However, as *Sox3* deletion in *Sox2*$^{+/+}$ ESCs did not increase *Sox1* and *Sox2* mRNA levels (*Figure 3D*) this suggests that the repressive effect of SOX3 on *Sox2* is sensitive to the SOX2 protein concentration.

Next, the ability of *Sox2*$^{fl/-}$; *Sox3*$^{-/Y}$ ESCs to transition to primed pluripotency was assessed by passaging in Activin/FGF. Whereas *Sox2*$^{fl/-}$ EpiSCs could be successfully established and maintained (*Figure 4*), *Sox2*$^{fl/-}$; *Sox3*$^{-/Y}$ cells displayed a differentiated morphology within two passages. Quantitative transcript analysis showed that in comparison to *Sox2*$^{fl/-}$ cells, *Sox2*$^{fl/-}$; *Sox3*$^{-/Y}$ cells induced less *Fgf5* but more Mash1 and *Pax6* (*Figure 8C*). These data suggest that while primed pluripotency can be maintained in the absence of either *Sox2* or *Sox3*, the transition from a naïve ESC state to a primed EpiSC state does not occur effectively when *Sox3* is absent and the *Sox2* gene dosage is halved. Notably, compared to *Sox2*$^{fl/-}$ ESCs, *Sox2*$^{fl/-}$; *Sox3*$^{-/Y}$ ESCs have the same *Sox2* mRNA level as *Sox2*$^{+/+}$ ESCs and *Sox1* mRNA is increased >10 fold (*Figure 8B*). Such an increase in *Sox1* mRNA expression is sufficient to enforce neural differentiation following LIF withdrawal (*Zhao et al., 2004*).

The increased levels of *Sox1* and *Sox2* mRNAs in LIF/FCS (*Figure 8B*), together with the increased levels of neural differentiation markers during EpiSC induction of *Sox2*$^{fl/-}$; *Sox3*$^{-/Y}$ ESCs (*Figure 8C*), prompted us to examine the behaviour of *Sox2*$^{fl/-}$; *Sox3*$^{-/Y}$ ESCs during neural differentiation. While *Sox2*$^{+/-}$ ESCs did not effectively undergo neural differentiation (*Figure 7*), deletion of *Sox3* from *Sox2*$^{fl/-}$ ESCs was sufficient to rescue neural differentiation as judged by induction of *Sox1*, Mash1 and *Pax6* mRNAs and of the βIII-tubulin protein (*Figure 8D*, *Figure 8—figure supplement 1*). This is likely a secondary consequence of the increase in *Sox2/Sox1* mRNA expression in *Sox2*$^{fl/-}$ ESCs resulting from *Sox3* deletion (*Figure 8B*). The two *Sox2*$^{+/+}$; *Sox3*$^{-/Y}$ ESC clones (*Figure 3A*) were able to differentiate into neural cells with similar efficiency to parental cells as shown by induction of *Sox1*, Mash1 and *Pax6* mRNAs (*Figure 8—figure supplement 2*) and appearance of βIII-tubulin-positive cells (*Figure 8—figure supplement 1*).

These data demonstrate that altering the SoxB1 genetic composition by deletion of *Sox3* and elimination of one functional *Sox2* allele induces a transcriptional de-regulation of the remaining

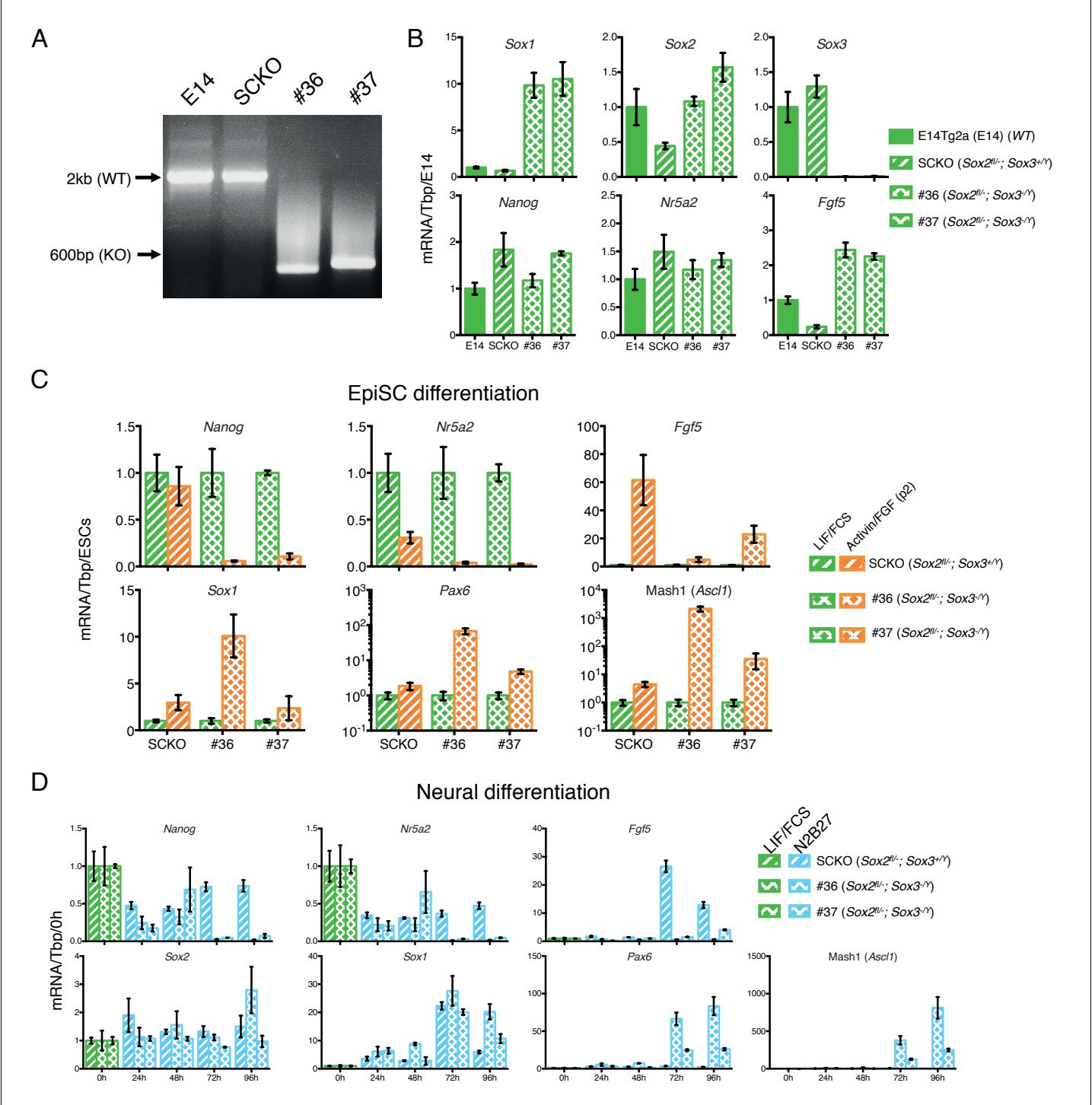

**Figure 8.** *Sox2/Sox3* requirements during EpiSCs differentiation. (**A**) Genotyping analysis of the *Sox3* locus in E14Tg2a (E14, *Sox2*^+/+^), SCKO (*Sox2*^fl/-^) ESCs (~2 kb band) and in two *Sox2*^fl/-^; *Sox3*^-/Y^ ESC clones (#36 and #37) derived after deletion of the *Sox3* locus (~600 bp band). Gene targeting was performed following the strategy depicted in *Figure 3A*. (**B**) RT-qPCR analysis of the indicated transcripts in E14Tg2a ESCs (E14, *Sox2*^+/+^), SCKO ESCs (*Sox2*^fl/-^) and in two *Sox2*^fl/-^; *Sox3*^-/Y^ ESC clones (#36 and #37). mRNA levels were normalised over *Tbp* and plotted relative to E14Tg2a. Error bars indicate the standard error of the mean (n = 3). (**C**) RT-qPCR analysis of the indicated transcripts in SCKO cells (*Sox2*^fl/-^) and in two *Sox2*^fl/-^; *Sox3*^-/Y^; *Sox2*^fl/-^ clones (#36 and #37) grown in LIF/FCS condition and in Activin/FGF conditions for 2 passages (P2). mRNA levels were normalised to *Tbp* and plotted relative to ESCs (LIF/FCS). Error bars indicate the standard error of the mean (n = 3). (**D**) RT-qPCR analysis of the indicated transcripts in SCKO cells (*Sox2*^fl/-^) and in two *Sox2*^fl/-^; *Sox3*^-/Y^ clones (#36 and #37) grown in LIF/FCS condition and in neural differentiation medium (N2B27) for the

*Figure 8 continued on next page*

Figure 8 continued

indicated number of hours. mRNA levels were normalised to *Tbp* and plotted relative to ESC (LIF/FCS). Error bars indicate the standard error of the mean (n = 3).
DOI: https://doi.org/10.7554/eLife.27746.022

The following figure supplements are available for figure 8:

**Figure supplement 1.** Immunofluorescence staining of the neural marker βIII-tubulin (TUJ1, green) in the indicated cells grown for 96 hr in neural differentiation medium (N2B27); DAPI staining represented in blue. Scale bar, 100 μm.
DOI: https://doi.org/10.7554/eLife.27746.023

**Figure supplement 2.** RT-qPCR analysis of the indicated transcripts in wild type E14Tg2a cells (*Sox2*$^{+/+}$; *Sox3*$^{+/Y}$) and in two *Sox2*$^{+/+}$; *Sox3*$^{-/Y}$ clones (S3KO8 and S3KO35) (see *Figure 3*) grown in LIF/FCS conditions and neural differentiation medium (N2B27) for the indicated number of hours.
DOI: https://doi.org/10.7554/eLife.27746.024

*SoxB1* alleles. The increased mRNA levels of *Sox1* and *Sox2* are then sufficient to rescue neural differentiation of *Sox2*$^{fl/-}$ ESCs but also impair the ability of *Sox2*$^{fl/-}$; *Sox3*$^{-/Y}$ ESCs to be captured as primed EpiSCs.

## Discussion

Our examination of SoxB1 function in pluripotent cells extends previous findings that SOXB1 proteins can act redundantly in ESCs (*Niwa et al., 2016*) and during somatic cell reprogramming (*Nakagawa et al., 2008*) by showing that in EpiSCs, SOX2 and SOX3 proteins are functionally redundant within an altered PGRN. Functional redundancy implies that we can consider the total SOXB1 complement as a key factor in the outcomes selected by cells genetically engineered to express varying combinations of functional *SoxB1* alleles. In particular, a broad range of SOXB1 levels are tolerated by ESCs. However, upon exit from naïve pluripotency, low SOXB1 levels permit entry to the EpiSC state, while high levels enforce neural differentiation (*Figure 9*). We have also uncovered cross-regulatory relationships between Sox3 and the other SoxB1 members, that we hypothesise act to maintain an adequate SOXB1 level.

### SOXB1 function in primed pluripotent cells is provided by SOX2 and SOX3

In ESCs *Sox2* mRNA is expressed at much higher levels than either *Sox1* or *Sox3*, and thus, in spite of redundancy, Sox2 can be considered to provide the dominant SOXB1 function in ESC pluripotency. However, even though *Sox2* is also the most abundant *SoxB1* transcript in EpiSCs, functional properties of the PGRN have changed. While *Sox2* mRNA levels in EpiSCs are lower than in ESCs, *Sox3* levels are increased, and SoxB1 function is provided redundantly by Sox2 and Sox3. *Sox1* mRNA levels are also increased in these cells. The ability of Sox1 or Sox3 to substitute functionally for Sox2 in ESCs might suggest that Sox1 could also function redundantly with Sox2 to maintain EpiSCs. However, Sox1 may be less relevant to pluripotency as, unlike Sox2 and Sox3, which are widely expressed in the pluripotent postimplantation epiblast, Sox1 expression is uniquely associated with neural fate in the epiblast (*Wood and Episkopou, 1999*; *Uchikawa et al., 2011*; *Cajal et al., 2012*; *Figure 9*). Indeed, Sox1 expression in EpiSC populations is associated with neural committed cells (*Tsakiridis et al., 2014*).

### SOXB1 redundancy in vivo

In the seminal *Sox2* deletion study (*Avilion et al., 2003*), *Sox2*$^{-/-}$ embryos fail to develop a postimplantation epiblast. This was hypothesised to be due to the fact that neither Sox1 nor Sox3 were expressed sufficiently at the time of embryonic failure and therefore no redundantly acting SOXB1 protein could compensate for the SOX2 absence. Additional instances of potential SoxB1 redundancy have been reported in vivo. Replacement of an endogenous *Sox2* allele with a Sox1 ORF produced no phenotype, suggestive of functional interchangeability (*Ekonomou et al., 2005*). SOXB1 redundancy is likely to be evolutionarily conserved since in the chick, Sox2 and Sox3 both promote development of ectoderm and neurectoderm at gastrulation, although interestingly, in this case Sox3 expression occurs before Sox2 (*Acloque et al., 2011*). Moreover, while genetic knock-ins have shown that placement of Sox2 ORF at the *Sox3* locus can rescue pituitary and testes phenotypes

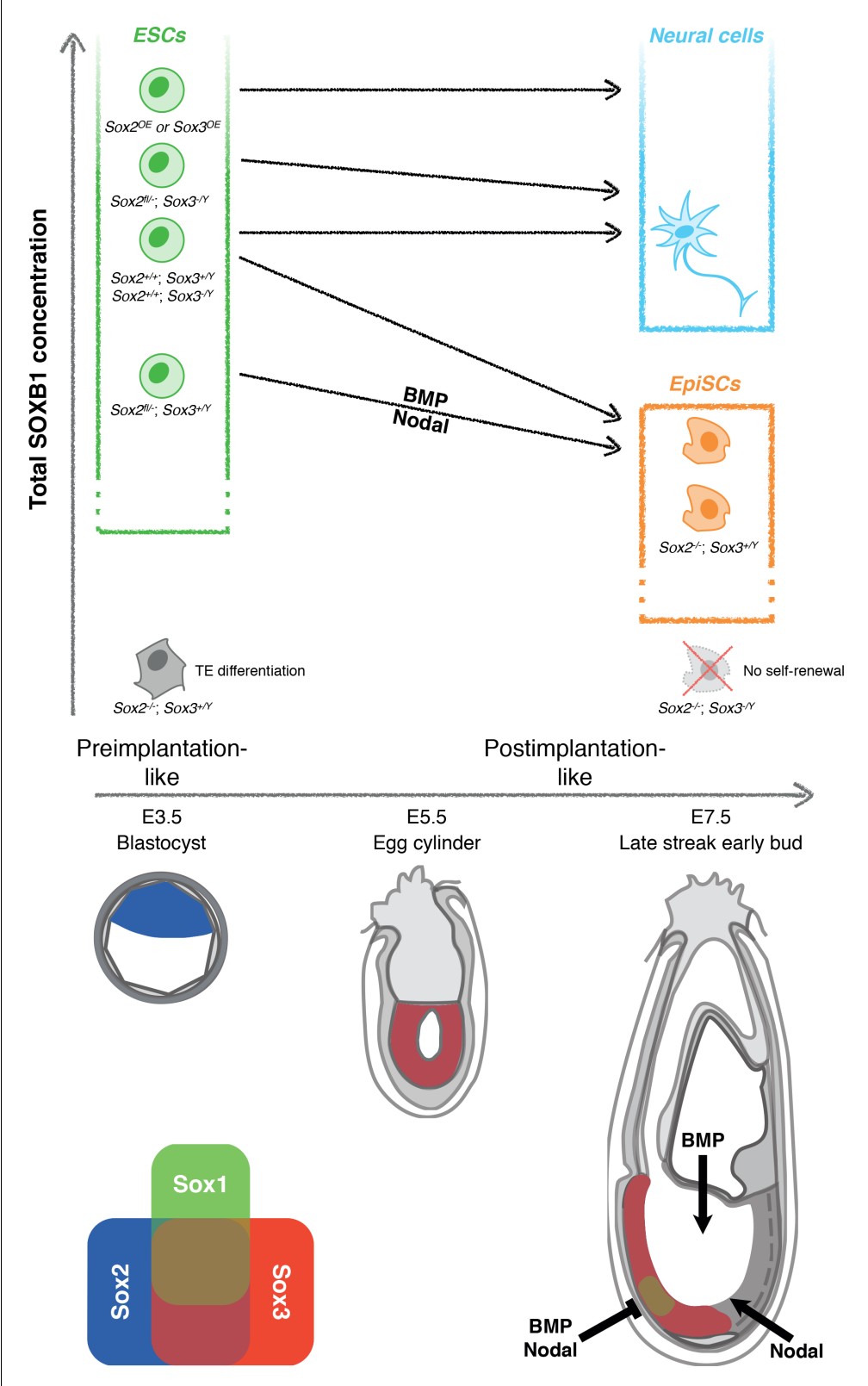

**Figure 9.** Dependence of cell fate potential of ESCs on the total SOXB1 concentration. The overall SOXB1 concentration inferred from SoxB1 transcript levels in different *SoxB1* mutant cell lines. Naïve ESCs self-renew in a wide range of SOXB1 concentrations. However, only ESCs with approximately wild-type SOXB1 levels can differentiate towards both primed EpiSCs and neural cells. ESCs with increased SOXB1 concentrations are poised towards neural differentiation, preventing their capture as primed EpiSCs in FGF/ActivinA. Decreased SOXB1 concentrations are insufficient to enable neural

*Figure 9 continued on next page*

Figure 9 continued

differentiation due to increased activity of neural antagonists (BMP and Nodal). A further reduction in SOXB1 is tolerated in the primed state due to SOX2/SOX3 functional redundancy but complete loss of the predominant SoxB1 forms is incompatible with self-renewal of both naïve and primed pluripotent cells. Depicted below are diagrams of pre- (E3.5) and postimplantation (E5.5 and E7.5) mouse embryos indicating the published expression patterns of SoxB1 mRNAs (*Wood and Episkopou, 1999*; *Uchikawa et al., 2011*; *Avilion et al., 2003*; *Cajal et al., 2012*), and the areas of BMP/Nodal signalling and inhibition during gastrulation (*Constam and Robertson, 2000*; *Bachiller et al., 2000*; *Kinder et al., 2001*; *Levine et al., 2006*; *Pereira et al., 2012*; *Norris et al., 2002*; *Lawson et al., 1999*; *Perea-Gomez et al., 2002*).

DOI: https://doi.org/10.7554/eLife.27746.025

caused by *Sox3* deletion (*Adikusuma et al., 2017*), the effects on pluripotent cells were not assessed. It is therefore interesting that while *Sox3$^{-/Y}$* mice can be viable, on a 129 genetic background they exhibit gastrulation-stage lethality (*Rizzoti and Lovell-Badge, 2007*; *Adikusuma et al., 2017*). This suggests that aspects of the regulation of SoxB1 expression that we show here to be important in vitro could also contribute to strain-specific differences in the timing or regulation of SOXB1 activity in vivo. Further studies will be required to establish the extent to which SOXB1 proteins can substitute genetically for one another in pluripotent cells in vivo.

## Low SOXB1 levels permit entry to the EpiSC state

Our results indicate that the total SoxB1 transcript level in ESCs needs to be reduced to enable entry into the EpiSC state rather than neural differentiation (*Figure 9*). *Sox2$^{+/-}$* ESCs placed in a neural differentiation protocol showed a reduced down-regulation of pluripotency markers (*Nanog* and *Nr5a2*) and failed to induce markers of neural differentiation. Surprisingly however, deletion of *Sox3* from SCKO cells to produce *Sox2$^{fl/-}$*; *Sox3$^{-/Y}$* ESCs restored neural differentiation capacity (*Figure 8D*). Moreover, placement of *Sox2$^{fl/-}$*; *Sox3$^{-/Y}$* ESCs in an EpiSC differentiation protocol resulted in a skewing of differentiation towards a neural identity as indicated by loss of *Nanog*, *Nr5a2* and *Fgf5* transcripts, and increase in mRNAs for neural differentiation markers (*Figure 8C*). This paradoxical behaviour may be explained by the fact that both *Sox1* and *Sox2* mRNAs increase upon elimination of *Sox3* from *Sox2$^{fl/-}$* ESCs (*Figure 8B*). Interestingly, reciprocal repression of *Sox3* by SOX2 also occurs in EpiSCs (*Figure 4E*). Unravelling the regulatory relationships between *SoxB1* genes is a relevant point for future studies.

## High SOXB1 levels enforce neural differentiation

Previous results have shown that elevating SOX1 or SOX2 expression in differentiating ESCs promotes neural differentiation (*Zhao et al., 2004*). In the present study, we showed that ESCs with either elevated SOX2 or SOX3 self-renew efficiently. However, when these cells were placed in an EpiSC differentiation protocol, differentiation was skewed towards a neural identity despite the presence of FGF and neural-antagonising Activin signals (*Vallier et al., 2004*, *2009*). This indicates that enforced expression of SOXB1 proteins overrides the signalling system that captures primed pluripotent cells in vitro.

## Parallels between SOXB1 function during in vitro and in vivo neurogenesis

The dynamics of ESC differentiation in vitro towards neural and EpiSC states show interesting parallels with early development in vivo (*Figure 9*). While ESCs and pre-implantation embryos express only *Sox2*, early post-implantation embryos, and differentiating ESCs, express *Sox2* and *Sox3* but not *Sox1* (*Wood and Episkopou, 1999*; *Uchikawa et al., 2011*; *Cajal et al., 2012*). At gastrulation stages, which are transcriptomically similar to EpiSCs (*Kojima et al., 2014*), *Sox2* and *Sox3* are expressed widely in the epiblast. As noted above, *Sox1* expression is restricted to a subdomain within the *Sox2*/*Sox3*-positive epiblast that overlaps extensively with the prospective brain (*Wood and Episkopou, 1999*; *Cajal et al., 2012*), where early neural differentiation occurs in vivo. Thus, the region expressing all three SOXB1 members might undergo neural induction as a consequence of expressing high levels of SOXB1 proteins, while the rest of the epiblast experiences lower SOXB1 levels, enabling pluripotency to extend through gastrulation (*Osorno et al., 2012*).

## An interplay between SOXB1 function and anti-neural signals

Inhibition of both BMP and Nodal signalling rescues the neural differentiation ability of $Sox2^{+/-}$ ESCs. These results suggest that $Sox2^{+/-}$ ESCs can initiate exit from naïve pluripotency but cannot complete neural differentiation due to enhanced responses of cells to endogenous anti-neuralising signalling by BMP and Nodal. The viability of $Sox2^{+/-}$ mice (*Avilion et al., 2003*; *Rizzoti and Lovell-Badge, 2007*; *Favaro et al., 2009*) indicates that the in vivo environment is able to overcome these anti-neural signals. In wild-type embryos, the prospective brain is shielded from Nodal and BMP signalling by secreted inhibitors of these pathways, including Cer1, Lefty1/2 (*Perea-Gomez et al., 2002*), Chrd and Nog (*Bachiller et al., 2000*) (*Figure 9*). Removal of either Nodal or BMP inhibition leads to absence of anterior neural tissue. Interestingly, the cells that express these inhibitors (including the anterior visceral endoderm and node) do not express SOXB1 proteins and are therefore likely to be functionally unaltered in $Sox2^{+/-}$ embryos. The observation that the reduced SOXB1 concentration in $Sox2^{+/-}$ pluripotent cells is compatible with neural differentiation, provided that endogenous anti-neuralising signals are blocked, indicates that to fully understand how the choice between neural and primed pluripotency is made, it will be necessary to elucidate how the signalling environment connects to the SOXB1-driven transcriptional programme.

# Materials and methods

## Cell culture

For a complete list of cell lines, their name, their genotypes and their original characterisation, see *Supplementary file 2*. All the cell lines used in this study were regularly tested for contaminations and were mycoplasma negative.

ESCs grown in LIF/FCS conditions were cultured on dishes coated with 1% gelatin (Sigma-Aldrich, St. Louis, USA) and in GMEM medium (Sigma-Aldrich, St. Louis, USA) supplemented with 1x non-essential aminoacids (Life Technologies, Waltham, USA), 1 mM sodium pyruvate (Life Technologies, Waltham, USA), 2 mM glutamine (Life Technologies, Waltham, USA), 100 U/ml human LIF (*Nichols et al., 1990*), 10% ESC-grade FCS (APS, UK) and 100 µM β-mercaptoethanol (Life Technologies, Waltham, USA). G418 (200 µg/mL, Sigma-Aldrich, St. Louis, USA) was supplemented to maintain SCKO ESCs.

To adapt ESCs into LIF/2i/N2B27 (*Ying et al., 2008*) (LIF/2i) or LIF/BMP4/N2B27 (*Ying et al., 2003b*) (LIF/BMP) conditions, ESCs were replated in LIF/FCS on gelatin-coated plates for 24 hr before changing the culture media to N2B27 medium (*Ying and Smith, 2003*) supplemented with 100 U/ml LIF, 1 µM PD0325901 (Stemgent, Cambridge, USA) and 3 µM CHIR99021 (Stemgent, Cambridge, USA) (LIF/2i) or 10 ng/ml BMP4 (Life Technologies, Waltham, USA) (LIF/BMP). Cells were passaged for at least four passages before using for analysis.

EpiSCs were derived from LIF/FCS-cultured ESCs as described previously (*Guo et al., 2009*; *Osorno et al., 2012*) and cultured on dishes pre-coated with 7.5 µg/ml fibronectin (Sigma-Aldrich, St. Louis, USA) without feeders in N2B27 medium (*Ying and Smith, 2003*) ActivinA (20 ng/ml, Peprotech, UK), bFGF (10 ng/ml, Peprotech, UK). In brief, ESCs were plated at a density of $3 \times 10^3$ cells/cm2; the equivalent of $30 \times 10^3$ cells per well of a six well plate. Medium was changed to Activin/FGF conditions 24 hr after replating. Cells were passaged after 4–5 days in a 1:20 dilution for the first 8–10 passages. Stable EpiSC lines were propagated by passaging in a 1:10-1:20 dilution.

For neural differentiation, ESCs were cultured with minor modifications as previously described (*Ying et al., 2003a*; *Ying and Smith, 2003*). Briefly, ESCs were replated in LIF/FCS on gelatin-coated plates for 24 hr before changing the culture media to N2B27 medium (*Ying and Smith, 2003*) only and allowed to grow for the indicated amount of time. When indicated, cells were differentiated in the presence of bFGF (10 ng/ml, Peprotech, UK), LDN-193189 (100 nM, Stemgent, Cambridge, USA) and/or SB-431542 (10 µM, Merck, Germany).

## *Sox2* deletion in ESCs and EpiSCs

Sox2 deletion in SCKO ESCs was performed similarly to previously described (*Gagliardi et al., 2013*; *Favaro et al., 2009*). In brief, $10^7$ SCKO ESCs grown in LIF/FCS conditions were transfected using Lipofectamine 3000 (Life Technologies, Waltham, USA) with 6–15 µg of transgene-expressing plasmid the indicated test cDNA before replated in LIF/FCS at a density of $1.5 \times 10^6$ per 10 cm dish

cultured overnight in the presence or in the absence of 4-hydroxytamoxifen (1 µM, Sigma-Aldrich, St. Louis, USA). Medium was changed 12 to 24 hr later to LIF/FCS medium supplemented with Hygromycin B (150 µg/mL, Roche, Switzerland). Transfected cells were cultured for 8 to 10 days before they were stained for alkaline phosphatase (AP) activity (Sigma-Aldrich, St. Louis, USA). Stained colonies were scored based on the presence of AP-positive cells within the colony. In parallel, unstained populations were expanded to generate cell lines before genotyping by PCR.

SCKO EpiSCs were transfected with ptdTomato-T2A-Cre by lipofection (Lipofectamine 2000, Life Technologies, Waltham, USA). After 12–24 hours cells were sorted for tdTomato expression and replated in the presence of ROCK inhibitor (Y-27632, Merck, Germany) for 24 hr before medium was changed to remove ROCK inhibitor. Clones were expanded before genotyping by PCR.

## CRISPR/Cas9 deletion of the Sox3 gene

Two sgRNAs were designed upstream of the ATG (sgRNA1) and downstream of the stop codon (sgRNA2) using an online CRISPR Design Tool (http://crispr.mit.edu/) and subsequently cloned in the BbsI-linearised pSpCas9(BB)−2A-GFP plasmid (Addgene 48138) as previously described (*Ran et al., 2013*). sgRNA sequences used in this study are listed in *Supplementary file 3*. $10^6$ E14Tg2a ESCs were co-transfected with 1 µg of sgRNA1-encoding plasmid and 1 µg of sgRNA2-encoding plasmid using Lipofectamine 3000 (Life Technologies, Waltham, USA) following the manufacturer's instructions. 24 hr after transfection, GFP-positive cells were sorted by FACS and replated at clonal density to allow the isolation of single colonies. After 8–10 days, single ESC colonies were expanded and the Sox3 locus was genotyped by PCR.

## Indel induction and TIDE analysis

SgRNAs targeting the Sox2 or Sox3 ORF immediately upstream of the sequence encoding for the HMG box were designed using an online CRISPR Design Tool (http://crispr.mit.edu/). The Sox2 sgRNA was subsequently cloned in the BbsI-linearised pSpCas9(BB)−2A-Puro (PX459) V2.0 plasmid (Addgene 62988) as previously described (*Ran et al., 2013*). The resulting plasmid, or the empty vector control, were then transfected using Lipofectamine 3000 (Life Technologies, Waltham, USA) into SCKO ESCs constitutively expressing either SOX1, SOX3 or GFP. After 24 hr, ESCs were selected with 1.5 µg/ml of puromycin (Life Technologies, Waltham, USA) for 24 hr to enrich for transfected cells, and then expanded for 72 hr. For Sox3 indel analysis, two individual sgRNAs were cloned in BbsI-linearised pSpCas9(BB)−2A-mCherry that was obtained by fusing a 2A-mCherry cassette to the Cas9 CDS of the eSpCas9(1.1) plasmid (Addgene 71814). $10^6$ E14Tg2a ($Sox2^{+/+}$), SCKO ($Sox2^{fl/-}$), SKO1 ($Sox2^{-/-}$) and SKO6 ($Sox2^{-/-}$) EpiSCs were then transfected with 1 µg of sgRNA-containing plasmids or empty vector using Lipofectamine 3000 (Life Technologies, Waltham, USA). 48 hr after transfection, mCherry-positive cells were FACS sorted, replated and expanded for 72 hr. Genomic DNA (gDNA) was extracted using the DNeasy Blood and Tissue kit (Qiagen, Germany) following the manufacturer's instructions. gDNA was PCR amplified using the Q5 HotStart Polymerase (NEB, Ipswich, USA) and primers flanking either the Sox2 sgRNA or Sox3 sgRNAs recognition sites. Primers and sgRNAs used in this study are listed in *Supplementary file 3*. PCR amplicons were purified and submitted for Sanger sequencing using the same forward primer that they had been generated with. Sanger sequencing electropherograms were then submitted for indel analysis with the TIDE tool (https://tide-calculator.nki.nl/, [*Brinkman et al., 2014*]) using amplicons obtained from cells transfected with empty vector as reference sequences. Indel analysis was performed using the default TIDE settings and a window of 27 bp for SCKO ESCs and Sox2 indel induction, and a window of 30 bp for EpiSCs and Sox3 indel induction. Indels with p value >0.001 were scored as statistically significant.

## PCR genotyping

Genomic DNA (gDNA) was extracted from cells using DNeasy kits (Qiagen, Germany) according to the manufacturer's recommended protocol. 100 ng of gDNA was used per PCR reaction. Primers used in this study are listed in *Supplementary file 3*.

## Plasmid constructs

A mouse genomic BAC containing 5 kb on either side of the *Sox2* stop codon (Source bioscience bMQ314D22) was shredded by pSC101-BAD-gbaA mediated recombineering using the pACYC177 backbone comprising the p15-origin and β-lactamase gene to produce pSox2-10kb. To construct the Sox2-T2A-H2B-tdTomato-IRES-Neo cassette, 89 bp and 168 bp immediately upstream and downstream of, and excluding the *Sox2* stop codon were PCR amplified to serve as homology arms for recombineering. The 89 bp upstream fragment has a 5' *Xho* I site and a GSG-T2A sequence at the 3' end containing an in-frame *Fse* I site that preserves the T2A Gly-Pro residues and is followed by *Not* I, *Pac* I, *Asc* I, *San* DI, *Bam* HI, *Afl* II, *Nhe* I, *Cla* I and *Xho* I sites. The sequence of this fusion is shown below:

GGCTCCGGAGAGGGCAGAGGAAGTCTGCTAACATGCGGTGACGTCGAGGAGAATCC
TGGGCCGGCCGCGGCCGCTTAATTAAGGCGCGCCGGGACCCGGATCCGCTTAAGGCTAGCA
TCGATTCTCGAG

The 89 bp fragment was cloned into a PCR amplified, pUC19-derived 2 kb minimal vector comprising β-lactamase gene and Col E1 origin with 150nt flanking sequences and a single *Xho* I site (pL). Individual features were PCR-amplified, flanked with unique sites and cloned as follows: H2B-tdTomato was cloned in-frame between *Fse* I and *Not* I, using TAA stop codon from *Pac* I; Gtx-IRES was cloned between *Pac* I and *Asc* I; neomycin phosphotransferase (Npt) was cloned between *San* DI and *Bam* HI, using the TAA stop codon from *Afl* II. The 168 bp downstream fragment is flanked with *Nhe* I at the 5' side and *Cla* I at the 3' end and was cloned between *Nhe* I and *Cla* I (pL-Sx-5HTiN3).

To insert a selection cassette for use in recombineering and targeting, a linker was made by annealing the following two oligonucleotides:

TCGAGCTTAAGGTCGACAGATCTCGATCGGCTAGCC
TCGAGGCTAGCCGATCGAGATCTGTCGACCTTAAGC

The linker was cloned into the *Xho* I site of a pTOPO-BluntII-derived, Zeocin-resistant version of pL (pZ-Linker). A 3.6 kb *Bam* HI fragment containing PGK-EM7-Npt-pA and MC1-HSVtk-pA flanked by FRT sites (FNF) was subcloned from pBS-M179 (a kind gift from Dr. Andrew Smith) into the *Bgl* II site of pZ-Linker (pZ-FNF), destroying the *Bam* HI and *Bgl* II sites. This places an *Afl* II site on one side and *Nhe* I site on the other side of the FNF cassette and these are used to subclone into pL-Sx-5HTiN3 to make pL-Sx-5HTiNFNF3.

To make the targeting vector, a 7.1 kb *Xho* I-*Cla* I fragment from pL-Sx-5HTiNFNF3 was transfected into *E. coli* containing pSox2-10kb and pSC101-BAD-gbaA to replace the *Sox2* stop codon by recombineering. Successful kanamycin-resistant recombinants carrying a 20 kb targeting vector (pSox2AHTiN-FNF-10kb) were amplified at 37°C to restrict pSC101-BAD-gbaA replication and identified by diagnostic *Bam* HI digest.

The targeting construct was linearised, electroporated into E14Tg2a ESCs and G418 resistant colonies expanded and genotyped by southern blot analysis as described below. The FRT-flanked cassette was removed by transiently transfecting pPGK-FlpO into verified E14TG2a-derived Sox2-tdTomato ESC clone 18 (TST18).

Overexpression constructs were generated by cloning open reading frames (ORFs) of indicated Sox genes, preceeded upstream by the Kozak consensus sequence (GCCGCCACC), into pPyCAG-IRES-Hyg vector between the *Xho*I and *Not*I sites (*Chambers et al., 2003*).

## Southern blot analysis

40 µg of genomic DNA were digested with *Eco* RI (5' probe analysis) or with *Hind* III (3' and internal probe analysis) and assessed by Southern blot analysis using probes synthesised by PCR from the oligonucleotides indicated in *Supplementary file 3*.

## RNA level quantification

Total RNA was purified from cells using RNeasy mini kits (Qiagen, Germany) following the manufacturer's protocol. cDNA was prepared from 1 µg of total RNA using SuperScript III reverse transcription kits (Life Technologies, Waltham, USA) according to the recommended protocol and the final cDNA solution was diluted 1:10 prior to use. 2 µl of cDNA solution was used per reaction with the

Takyon Sybr Assay (Eurogentec, Belgium). Primers used in this study are listed in *Supplementary file 3*.

## Microarray gene expression analysis

Total RNA (127 ng/sample) from three independently replicated experiments was converted into biotin-labelled cRNA using the Illumina TotalPrep RNA amplification kit (Ambion, Cambridge, USA). Microarray hybridization reactions were performed on a Mouse WG-6v2 BeadChip (Illumina). Raw data were normalised in R using the beadarray (*Dunning et al., 2007*), limma (*Smyth, 2005*) and sva (*Leek and Storey, 2007*) packages from the Bioconductor suite (*Gentleman et al., 2004*). Briefly, low-quality probes were removed from the input and data were then quantile-normalized. ComBat was used to account for batch effects (*Leek and Storey, 2007*) between microarrays run at different dates. Differential expression in the log2-transformed data was assessed with the limma algorithms (*Smyth, 2005*). Probes were considered differentially expressed if they showed a FDR-adjusted p-value of $\leq$ 0.1 and an absolute $\log_2$ fold change $\geq$ log2(1.5). Primary analysis results were uploaded to GeneProf (*Halbritter et al., 2011*) and mapped to Ensembl-based reference genes, collapsing multiple probes for the same gene by picking the most responsive probe (i.e., the probe with the highest absolute fold change across all pair-wise comparisons). Data are available in *Supplementary file 1*. Microarray data have been submitted to the Gene Expression Omnibus (GEO) under accession code GSE99185.

## Immunofluorescence staining

Cells were fixed with 4% PFA (10 mins, RT) and permeabilised using PBS/0.1% (v/v) Triton-X100 (PBSTr) for 10 mins, before quenching with 0.5M Glycine/PBSTr (15 mins). Non-specific antigens were blocked using 3% (v/v) donkey serum/1% (v/v) BSA/PBSTr (1 hr, RT) before incubating with primary antibody in blocking solution (4°C, overnight). Cells were washed with PBSTr before incubating with donkey-raised secondary antibodies conjugated with Alexa-488,–568 or $-$647 in blocking solution (1–2 hr, RT). DAPI (1–2 µg/mL, Molecular Probes) in PBS was added to cells for at least 30 mins before imaged using the Olympus IX51 inverted fluorescent microscope. Primary antibodies used were: $\alpha$-SOX2 (1:400; Abcam ab92494, UK ), $\alpha$-NANOG (1:500; eBioscience 14–5761, Cambridge, USA) and $\alpha$-OCT4 (1:400; Santa Cruz Biotechnology sc8628, Santa Cruz, USA), $\alpha$-SOX3 antibody (1:400; Abcam ab42471, UK ), $\alpha$-βIII-tubulin (1:1500; Covance MMS-435P, Princeton, USA ).

## Immunoblotting analysis

Cells were lysed with lysis buffer comprising 50 mM Tris pH 8.0, 150 mM NaCl supplemented with fresh 0.5% NP-40, 0.5 mM DTT, 1 $\times$ protease inhibitors cocktail (Roche, Switzerland) and 1.3 µl of Benzonase (Novagen, Germany) (1 hr, 4°C). Samples were prepared by boiling 40 µg of total protein extract with Laemmli buffer (Life Technologies, Cambridge, USA) . Samples were analysed using Bolt 10% Bis-Tris +SDS PAGE (Life Technologies, Cambridge, USA) and electroblotted onto 0.2 µm pore Whatman-Protran nitrocellulose membranes (Capitol Scientific, Austin, USA) in transfer buffer comprising 25 mM Tris/0.21M glycine/20% methanol. Membranes were blocked using 5% (w/v) low-fat milk in 0.01% (v/v) Tween-20/PBS (PBSTw) before incubating with primary antibody in blocking solution . Membranes were washed with PBSTw before incubating with donkey-raised secondary antibodies conjugated with IRDye 800CW (1:10000; LI-COR 926–32213, Lincoln, USA) and HRP-conjugated $\alpha$-βActin (1:10000; Abcam ab20272, UK) antibody. HRP-staining was developed using a Super-signal West Pico kit (Pierce, Cambridge, USA) before imaging the membranes using LI-COR Odyssey Fc imager. The primary antibodies used were: $\alpha$-Sox2 (1:1000; Abcam ab92494, UK) and $\alpha$-Nanog (1:2000; Bethyl Laboratories A300-397A, Montgomery, USA).

## Embryo manipulation

Mice were maintained on a 12 hr light/dark cycle. All animals were maintained and treated in accordance with guidance from the UK Home Office. Embryonic day (E)0.5 was designated as noon on the day of finding a vaginal plug. Morula aggregations, blastocyst injections and embryo transfer were performed using standard procedures. Chimeric, gastrulation-stage embryos were collected at E7.5 and imaged using an inverted fluorescence microscopy. Chimeric E9.5 embryos were fixed with

4% PFA at 4°C for 3 hr and cryosectioned as described before (*Wymeersch et al., 2016*). Sections were stained for GFP as described above.

## Acknowledgements

We thank Emily Erickson and Christine Watson (Cambridge) for suggesting the use of TIDE analysis; CRM Animal House, Microscopy Facility and FACS Facility staff for assistance; Donal O'Carroll for comments on the manuscript. This research is supported by grants from the Medical Research Council (IC and VW) and the Biotechnological and Biological Sciences Research Council of the United Kingdom (IC) and by a Japan Partnering Award from the Biotechnological and Biological Sciences Research Council (IC). FCKW was supported by an MRC studentship and a Centenary Award; EM received an award from the International Association for the Exchange of Students for Technical Experience.

## Additional information

### Funding

| Funder | Grant reference number | Author |
| --- | --- | --- |
| Medical Research Council | | Andrea Corsinotti<br>Frederick CK Wong<br>Florian Halbritter<br>Douglas Colby<br>Nicholas P Mullin<br>Valerie Wilson<br>Ian Chambers |
| Biotechnology and Biological Sciences Research Council | | Elisa Hall-Ponsele<br>Ian Chambers |
| Biotechnology and Biological Sciences Research Council | Japan partnering award | Ian Chambers |

The funders had no role in study design, data collection and interpretation, or the decision to submit the work for publication.

### Author contributions

Andrea Corsinotti, Conceptualization, Data curation, Formal analysis, Supervision, Validation, Investigation, Visualization, Methodology, Writing—original draft, Writing—review and editing; Frederick CK Wong, Conceptualization, Data curation, Formal analysis, Supervision, Validation, Investigation, Visualization, Methodology, Writing—review and editing; Tülin Tatar, Investigation, Performed differentiation and genotyping analyses; Iwona Szczerbinska, Kirsten Liggat, Investigation, Performed differentiation analyses; Florian Halbritter, Data curation, Formal analysis, Investigation, Visualization, Methodology, Performed bioinformatic analyses; Douglas Colby, Investigation, Performed cell culture analyses; Sabine Gogolok, Methodology, Assisted with TIDE analysis; Raphaël Pantier, Elisa Hall-Ponsele, Investigation, Performed protein expression analyses; Elham S Mirfazeli, Investigation, Initiated TIDE analyses; Nicholas P Mullin, Investigation, Performed gene expression analyses; Valerie Wilson, Ian Chambers, Conceptualization, Data curation, Formal analysis, Supervision, Funding acquisition, Investigation, Visualization, Writing—original draft, Project administration, Writing—review and editing

### Author ORCIDs

Andrea Corsinotti (iD) http://orcid.org/0000-0003-4481-0999
Florian Halbritter (iD) https://orcid.org/0000-0003-2452-4784
Valerie Wilson (iD) http://orcid.org/0000-0003-4182-5159
Ian Chambers (iD) http://orcid.org/0000-0003-2605-1597

### Ethics

Animal experimentation: Animal experiments were performed under the UK Home Office project license PPL60/4435, approved by the Animal Welfare and Ethical Review Panel of the MRC Centre for Regenerative Medicine and within the conditions of the Animals (Scientific Procedures) Act 1986.

### Decision letter and Author response

Decision letter https://doi.org/10.7554/eLife.27746.033
Author response https://doi.org/10.7554/eLife.27746.034

## Additional files

### Supplementary files

• Supplementary file 1. Microarray gene expression data described in *Figure 1*.
DOI: https://doi.org/10.7554/eLife.27746.026

• Supplementary file 2. List of the cell lines used in this study with their genotype and additional transgenes if present.
DOI: https://doi.org/10.7554/eLife.27746.027

• Supplementary file 3. List of the PCR primers and sgRNAs used in this study.
DOI: https://doi.org/10.7554/eLife.27746.028

• Transparent reporting form
DOI: https://doi.org/10.7554/eLife.27746.029

### Major datasets

The following dataset was generated:

| Author(s) | Year | Dataset title | Dataset URL | Database, license, and accessibility information |
| --- | --- | --- | --- | --- |
| Corsinotti A, Wong FCK, Tatar T, Szczerbinska I, Halbritter F, Colby D, Gogolok S, Pantier R, Liggat K, Mirfazeli E, Hall-Ponsele E, Mullin N, Wilson V, Chambers I | 2017 | Distinct SoxB1 networks are required for naïve and primed pluripotency | https://www.ncbi.nlm.nih.gov/geo/query/acc.cgi?acc=GSE99185 | Publicly available at the NCBI Gene Expression Omnibus (accession no: GSE99185) |

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
