## [Decision Letter]

Thank you for submitting your article "Distinct SoxB1 networks are required for naive and primed pluripotency" for consideration by *eLife*. Your article has been reviewed by two peer reviewers, and the evaluation has been overseen by a Reviewing Editor and Sean Morrison as the Senior Editor. The reviewers have opted to remain anonymous.

The reviewers have discussed the reviews with one another and the Reviewing Editor has drafted this decision to help you prepare a revised submission.

Essential revisions:

All reviewers agree that the study has been carefully performed and contains some important findings about the role of Sox genes in pluripotency. However, there is much previous work pertinent to the study which is not cited in the manuscript. This omission should be addressed, and the relevant findings discussed in light of the authors' work. You should also discuss the ways in which in vivo findings are at odds with the in vitro findings and why you have confidence that the differences do not simply reflect in vitro artifacts.

Specific comments:

This aim of this work is to address the role of SOXB1 factors in epiblast stem cell (EpiSC) pluripotency. The authors show that Sox2 can be deleted from EpiSCs without affecting their potential, which is due to functional redundancy with Sox3. This is claimed to be a surprising result, but it is really quite predictable, given the co-expression of the two genes, and that they are known to be functionally equivalent (see for example: https://www.ncbi.nlm.nih.gov/pubmed/28515211;https://www.ncbi.nlm.nih.gov/pubmed/10446282), with Sox3 at slightly higher levels and slightly more extensively expressed than Sox2. Indeed, compared to the mouse, the two genes have swapped prominence in the chick embryo, with Sox3 preceding Sox2, and with both acting in the same way antagonistically to Snail, in helping to promote ectoderm/neurectoderm development during gastrulation (https://www.ncbi.nlm.nih.gov/pubmed/21920318).

The authors show that Sox3 can also be deleted from EpiSCs without eliminating self-renewal. Again, this is not surprising, given that Sox3 null mutant mice are viable (https://www.ncbi.nlm.nih.gov/pubmed/14981518; https://www.ncbi.nlm.nih.gov/pubmed/17728342; https://www.ncbi.nlm.nih.gov/pubmed/28515211).

Deletion of both Sox2 and Sox3 was found to prevent self-renewal of EpiSCs. This is again not surprising, but it is probably worth showing.

The authors also found that differentiation of Sox2 heterozygous ESCs is compromised, although this would seem not to be very important in vivo, given that such cells can make germ line chimeras after blastocyst injection and that both Sox2 heterozygous mice and humans are viable (although with a range of more or less severe defects) [many papers]. The authors also show that increased SOXB1 levels divert the ESC to EpiSC transition towards neural differentiation.

Putting all their findings together, they claim that optimal SOXB1 levels are critical for each pluripotent state and for cell fate decisions during exit from naïve pluripotency.

While I have few if any problems with the way the experiments have been conducted, indeed some approaches are clever, such as the indel analysis after CRISPR/Cas9 NHEJ mutagenesis, the authors seem to disregard much published data including all that on the effects of mutations in Sox2, Sox3 (and Sox1) in vivo, where there is extensive data both in mice and humans. Several of the in vivo findings are at odds with the claims being made here for the role of Soxb1 genes in ESCs and EpiSCs in vitro. This begs the question as to the relevance of this current work. If the authors had gone on to define the downstream genes required for neurectoderm, etc., fate, or the mechanisms involved in crosstalk or feedback regulation between Sox2 and Sox3, then this would be important. But as it is, I worry that they are mostly chasing some in vitro artifacts.

Other comments:

It is usual convention to write protein symbols as all uppercase letters (both mouse and human proteins).

Introduction, third paragraph: Where is the evidence for 23 SOX proteins in mammals? It is usually accepted that there are 20.

Subsection “A subset of SOX family proteins can functionally replace *Sox2* in ESC self-renewal”, first paragraph and Figure 2: Sox15 was already known to be able to substitute for Sox2 in pluripotency. https://www.ncbi.nlm.nih.gov/pubmed/27582319

Subsection “Sox3 is dispensable for pluripotent cell self-renewal”, first paragraph and Figure 3—figure supplement 1: The level of Sox3 in undifferentiated ES cells is extremely low – but it is readily activated by retinoic acid and other ligands.

"These findings indicate that Sox3 is dispensable for ESC self-renewal." This was also known from previous work from others where Sox3 null mutant ESCs have been generated (gene targeting via homologous recombination) and the generation of mice from these (https://www.ncbi.nlm.nih.gov/pubmed/14981518;https://www.ncbi.nlm.nih.gov/pubmed/14585968).

"These data demonstrate that both naïve and primed pluripotent cells can self-renew in the absence of Sox3." Again, this is not surprising given the viability of Sox3 null mice.

Subsection “*Sox2* is dispensable for the maintenance of primed pluripotency”, last paragraph: Sox3 mRNA levels were found to be elevated in in *Sox2*^-/-^ EpiSCs compared to controls. Again, this has been seen before in the developing brain in vivo (https://www.ncbi.nlm.nih.gov/pubmed/18638478).

"These data indicate that a Sox2 level above that present in *Sox2^fl/-^*ESCs is required to enable neural differentiation and suggest that below this *SOX2* concentration cells are directed towards non-neural differentiation via an early post-implantation identity (Figure 6)." This does not make sense since *Sox2*^+/-^ mice (and humans) do have a CNS, although with some abnormalities. Is the *Sox2^fl^* allele hypomorphic?

"While *Sox2^fl/-^*ESCs do not effectively undergo neural differentiation (Figure 6), quantitative transcript analysis shows that deletion of Sox3 from *Sox2^fl/-^*ESCs is sufficient to rescue the neural differentiation of the cells.." I don't understand this. And again, it doesn't relate very well to the in vivo situation.

– "In the future, it will also be interesting to determine the extent to which SoxB1 proteins can substitute for Sox2 function in vivo." Some such combinations have already been carried out:https://www.ncbi.nlm.nih.gov/pubmed/15882093https://www.ncbi.nlm.nih.gov/pubmed/28515211

Discussion, fourth paragraph: This paper, which also shows co-expression and some functional redundancy of Sox2 and Sox3, should perhaps also be cited in relevant places: https://www.ncbi.nlm.nih.gov/pubmed/12514105

---

## [Author Response]

Essential revisions:All reviewers agree that the study has been carefully performed and contains some important findings about the role of Sox genes in pluripotency. However, there is much previous work pertinent to the study which is not cited in the manuscript. This omission should be addressed, and the relevant findings discussed in light of the authors' work. You should also discuss the ways in which in vivo findings are at odds with the in vitro findings and why you have confidence that the differences do not simply reflect in vitro artifacts.

We are pleased that our reviewers agree that our work has been carefully conducted and has produced important results about the function of Sox genes in pluripotency. We acknowledge that we omitted discussion of several important and relevant previous publications and are grateful to the reviewers for drawing our attention to these omissions. As a consequence, we have considerably redrafted our Discussion to include this published work in the context of our findings. Moreover, in response to our reviewers prompting, we have now performed additional experiments that we think resolve the apparent discrepancy between published in vivo studies and our in vitro work. We believe our current manuscript has been significantly improved in response to our reviewers’ request and we hope that it is now considered suitable for publication.

Specific comments:This aim of this work is to address the role of SOXB1 factors in epiblast stem cell (EpiSC) pluripotency. The authors show that Sox2 can be deleted from EpiSCs without affecting their potential, which is due to functional redundancy with Sox3. This is claimed to be a surprising result, but it is really quite predictable, given the co-expression of the two genes, and that they are known to be functionally equivalent (see for example: https://www.ncbi.nlm.nih.gov/pubmed/28515211;https://www.ncbi.nlm.nih.gov/pubmed/10446282), with Sox3 at slightly higher levels and slightly more extensively expressed than Sox2.

We are grateful to the reviewer for pointing us to the Adikusuma et al. paper, an elegant study published while our manuscript was under review, which shows that knock-in of Sox2 to the Sox3 locus can rescue pituitary and testis defects of the Sox3 knockout. We refer to this work as indicated below (Introduction, last paragraph and subsection “SOXB1 redundancy in vivo”). The second paper referred to above is Wood and Episkopou, a classic RNA in situ hybridisation study that we have cited in our manuscript and that shows overlapping Sox2 and Sox3 expression domains in the post-implantation epiblast. We accept the reviewer’s point that these overlapping expression domains, together with other studies referred to later by the reviewer are sufficient to suggest our findings are unsurprising. Therefore, we have deleted the relevant instances of ‘surprising’ from our manuscript and reframed our discussion in light of the papers that the reviewer has kindly brought to our attention. However, neither of the studies above, nor any of the studies that the reviewer refers to below have tested the functional redundancy of Sox2 and Sox3 in postimplantation pluripotency and therefore our results are novel.

We have reserved the use of the word ‘surprising’ in the revised manuscript for the unexpected up-regulation of Sox2 mRNA upon deletion of Sox3, rescuing neural differentiation of *Sox2*^+/-^ cells and indicating a novel cross-regulatory relationship between the SoxB genes (subsection “Modulating SOXB1 levels affects differentiation and can prevent capture of primed pluripotency, fifth paragraph and subsection “Low SoxB1 levels permit entry to the EpiSC state”).

Indeed, compared to the mouse, the two genes have swapped prominence in the chick embryo, with Sox3 preceding Sox2, and with both acting in the same way antagonistically to Snail, in helping to promote ectoderm/neurectoderm development during gastrulation (https://www.ncbi.nlm.nih.gov/pubmed/21920318).

We agree this is relevant background and have included a reference to the Acloque paper in the Discussion, ‘in the chick, Sox2 and Sox3 both promote development of ectoderm and neurectoderm at gastrulation, although interestingly, in this case Sox3 expression occurs before Sox2 (Acloque et al.).’

The authors show that Sox3 can also be deleted from EpiSCs without eliminating self-renewal. Again, this is not surprising, given that Sox3 null mutant mice are viable (https://www.ncbi.nlm.nih.gov/pubmed/14981518; https://www.ncbi.nlm.nih.gov/pubmed/17728342; https://www.ncbi.nlm.nih.gov/pubmed/28515211).

The papers referred to are Rizzoti et al. (shows Sox3 is required for pituitary function), Rizzoti & Lovell-Badge (shows that Sox3 is required for correct craniofacial morphogenesis) and Adikusuma et al. (knock-in of Sox2 to the Sox3 locus referred to above). While each of these papers indicates that *Sox3*-null mice can be viable and that therefore there may not be an essential requirement for Sox3 in the epiblast, there are suggestions that things may not be so simple.

Rizzotti et al. (2004) state as an unpublished observation that there is early gastrulation lethality of Sox3 on 129 genetic background and that a viable phenotype was only observed in conditional nulls on an outbred background. Adikusuma et al. also report failure to generate Sox3^Sox2KI^ mice from targeted clones via conventional gene targeting in ES cells, followed by blastocyst injection of these cells. Subsequently, Adikusuma et al. obtained the *Sox3^Sox2KI^* allele using a CRISPR/Cas9 strategy via pronuclear injection of C57Bl/6 zygotes. The failure of two independent groups to establish mice from knock-out Sox3 on a 129 background points to a requirement for Sox3 that goes beyond the reported viability on an outbred background.

We have shown that Sox3 contributes to the total SoxB1 levels required for postimplantation pluripotency (Figure 5). This suggests that subtle strain-specific variations in Sox2 levels/timing in postimplantation embryos may exist, which permit maintenance of Sox3 KO ES cells but not embryos.

Deletion of both Sox2 and Sox3 was found to prevent self-renewal of EpiSCs. This is again not surprising, but it is probably worth showing.

We agree that despite other reports of tissues where Sox2 and Sox3 are redundant, our finding that Sox2 and Sox3 act redundantly in EpiSC self-renewal is worthwhile, especially as it highlights a distinct post-implantation-specific functional PGRN. Moreover, our results point to a model wherein both genes contribute to maintain threshold SoxB1 levels required for postimplantation pluripotency – as alluded to by Avilion et al. This information is contained in the updated model (Figure 9).

The authors also found that differentiation of Sox2 heterozygous ESCs is compromised, although this would seem not to be very important in vivo, given that such cells can make germ line chimeras after blastocyst injection and that both Sox2 heterozygous mice and humans are viable (although with a range of more or less severe defects) [many papers]. The authors also show that increased SOXB1 levels divert the ESC to EpiSC transition towards neural differentiation.Putting all their findings together, they claim that optimal SOXB1 levels are critical for each pluripotent state and for cell fate decisions during exit from naïve pluripotency.While I have few if any problems with the way the experiments have been conducted, indeed some approaches are clever, such as the indel analysis after CRISPR/Cas9 NHEJ mutagenesis, the authors seem to disregard much published data including all that on the effects of mutations in Sox2, Sox3 (and Sox1) in vivo, where there is extensive data both in mice and humans. Several of the in vivo findings are at odds with the claims being made here for the role of Soxb1 genes in ESCs and EpiSCs in vitro. This begs the question as to the relevance of this current work. If the authors had gone on to define the downstream genes required for neurectoderm, etc., fate, or the mechanisms involved in crosstalk or feedback regulation between Sox2 and Sox3, then this would be important. But as it is, I worry that they are mostly chasing some in vitro artifacts.

We apologise for omitting important in vivo studies in our previous manuscript. We have taken on board the reviewers’ suggestions and have now discussed previous in vivo findings in relation to our results (we have indicated these in response to detailed points elsewhere in this document).

To address whether we were indeed chasing an artefactual inability of SCKO to undergo neural differentiation, we independently generated *Sox2*^+/-^ ESCs using CRISPR. Two new *Sox2*^+/-^ ESC clones derived independently using two different sgRNAs confirmed our findings that mutation of a *Sox2* allele in ESCs causes a defect in neural differentiation (Figure 7, Figure 7—figure supplement 1). Whether this was due to Sox2 protein level being insufficient to drive neural differentiation remained an open question that we have addressed as outlined below.

in vitro findings can highlight cellular requirements that are difficult to unpick in vivo. A key difference between in vitro and in vivo experiments is the ability to control the extracellular environment. Through in vitro studies using inhibitors of BMP and Nodal signalling, we have uncovered endogenous BMP and Nodal signalling in Sox2 heterozygous cells that block neural differentiation (Figure 7). This establishes that the inability of *Sox2*^+/-^ ESCs to undergo neural differentiation is not an intrinsic cellular defect, but is due to an anti-neural environment. In the embryo, such an environment exists in the anterior epiblast and does not depend on Sox2. We believe these experiments resolve the apparent discrepancy between our findings and the ability of *Sox2*^+/-^ mice to develop brains.

Other comments:It is usual convention to write protein symbols as all uppercase letters (both mouse and human proteins).

We have corrected the text so that all reference to proteins is uppercase.

Introduction, third paragraph: Where is the evidence for 23 SOX proteins in mammals? It is usually accepted that there are 20.

We apologise for this error, which has been corrected (Introduction, third paragraph).

Subsection “A subset of SOX family proteins can functionally replace Sox2 in ESC self-renewal”, first paragraph and Figure 2: Sox15 was already known to be able to substitute for Sox2 in pluripotency. https://www.ncbi.nlm.nih.gov/pubmed/27582319

We have cited the Niwa et al. study but have added a further citation to this work relevant to SOX15 (subsection “A subset of SOX family proteins can functionally replace Sox2 in ESC self-renewal”, first paragraph).

Subsection “Sox3 is dispensable for pluripotent cell self-renewal”, first paragraph and Figure 3—figure supplement 1: The level of Sox3 in undifferentiated ES cells is extremely low – but it is readily activated by retinoic acid and other ligands.

We have amended the text (subsection “Sox3 is dispensable for both naïve and primed pluripotency”, first paragraph) to draw attention to the low Sox3 expression level observed in ESCs and to our intention to document whether this was required for ESC self-renewal.

"These findings indicate that Sox3 is dispensable for ESC self-renewal." This was also known from previous work from others where Sox3 null mutant ESCs have been generated (gene targeting via homologous recombination) and the generation of mice from these (https://www.ncbi.nlm.nih.gov/pubmed/14981518;https://www.ncbi.nlm.nih.gov/pubmed/14585968).

The papers referred to above are Rizzoti et al. and Weiss et al. Both papers document the existence of ESCs carrying *loxP*-flanked *Sox3* alleles but do not document deletion of either of these *loxP*-flanked *Sox3* alleles in ESCs. Rizzoti et al. do refer to unpublished data in which a null mutation in Sox3 was targeted to the *Sox3* allele in XY male ESCs. We have therefore amended our text to reflect this (“These findings indicate that Sox3 is dispensable for ESC self-renewal, confirming previously reported unpublished data (Rizzoti et al)”).

"These data demonstrate that both naïve and primed pluripotent cells can self-renew in the absence of Sox3." Again, this is not surprising given the viability of Sox3 null mice.

The reviewer raises an interesting point. Rizzoti et al. state that “To study the role of Sox3, we targeted null mutations in the gene into XY embryonic stem (ES) cells, but injection of these cells into blastocysts resulted in early lethality of the chimeras due to a gastrulation defect (unpublished data).” The authors circumvented this problem by generating a *loxP*-flanked *Sox3* allele and testing the effects of Sox3 deletion in an outbred background. Rizzoti et al.’s unpublished data suggests in fact that Sox3 may in fact be required in postimplantation pluripotent cells in specific genetic backgrounds (in this case, 129sv/ev). Interestingly, genetic background effects do modulate the severity of Sox3 phenotypes since *Sox3*-null mice have a more severe pituitary phenotype on a C57Bl/6 background (Adikusuma et al.) than on an MF1 background (Rizzoti et al.)

We discussed the points raised by these papers in the Discussion (subsection “SOXB1 redundancy in vivo”).

Subsection “Sox2 is dispensable for the maintenance of primed pluripotency”, last paragraph: Sox3 mRNA levels were found to be elevated in in Sox2^-/-^ EpiSCs compared to controls. Again, this has been seen before in the developing brain in vivo (https://www.ncbi.nlm.nih.gov/pubmed/18638478).

This paper by Miyagi et al. shows that Sox3 mRNA is increased in *Sox2*-null E14.5 brain, suggesting that functional compensation between Sox2 and Sox3 occurs in neural stem cells. We have included citation of the work of Miyagi et al. and have amended the text accordingly (subsection “Sox2 is dispensable for the maintenance of primed pluripotency”, last paragraph).

"These data indicate that a Sox2 level above that present in Sox2^fl/-^ ESCs is required to enable neural differentiation and suggest that below this SOX2 concentration cells are directed towards non-neural differentiation via an early post-implantation identity (Figure 6)." This does not make sense since Sox2^+/-^ mice (and humans) do have a CNS, although with some abnormalities.

We agree with the reviewers’ comment and have amended the text (subsection “Modulating SOXB1 levels affects differentiation and can prevent capture of primed pluripotency”, second paragraph) to underline that our findings are referring to the differentiation potential of ESCs in vitro, and that this does not exclude compensation mechanisms in vivo. As mentioned above, we have now investigated the neural differentiation phenotype further and believe we can suggest an explanation for this discrepancy based on their increased BMP/Nodal signalling activity.

Is the Sox2^fl^ allele hypomorphic?

mRNA and protein expression analysis in this (Figure 3—figure supplement 1, Figure 4, Figure 8) and previous studies (Gagliardi et al. 2013) suggests that *Sox2*^+^/- cells express about 50-60% of the Sox2 mRNA/protein levels found in WT cells. The validation of the neural differentiation defect of *Sox2*^+/-^ ESCs using ESC clones derived by introducing small deletions in the Sox2 CDS by CRISPR indicates that this defect is not peculiar to the SCKO ESC line.

"While Sox2^fl/-^ ESCs do not effectively undergo neural differentiation (Figure 6), quantitative transcript analysis shows that deletion of Sox3 from Sox2^fl/-^ ESCs is sufficient to rescue the neural differentiation of the cells.." I don't understand this. And again, it doesn't relate very well to the in vivo situation.

We have dealt with the apparent differences between the Sox2 heterozygote phenotype in vitro and in vivo above. To clarify the statement, we have added text (subsection “Modulating SOXB1 levels affects differentiation and can prevent capture of primed pluripotency”). This indicates that removal of Sox3 from *Sox2^fl/-^*ESCs increases Sox2 expression to wild-type levels, and Sox1 expression to 10x the wild-type level. This could explain why *Sox2^fl/-^*Sox3null ESCs can differentiate into neurons. This effect is rather surprising given the low expression level of Sox3 relative to *Sox2* in ESCs. This suggests that small alterations in feedback between SoxB1 genes may be sufficient to rescue neural differentiation.

"In the future, it will also be interesting to determine the extent to which SoxB1 proteins can substitute for Sox2 function in vivo." Some such combinations have already been carried out: https://www.ncbi.nlm.nih.gov/pubmed/15882093https://www.ncbi.nlm.nih.gov/pubmed/28515211

These papers refer to a Sox1knock-in at Sox2 (Ekonomou et al.) and the Sox2 knock-in at Sox3 mentioned earlier (Adikusuma et al.). We have discussed these findings in the “Discussion” section “SOXB1 redundancy in vivo”, ending with a modified version of the sentence above; “Further studies will be required to establish the extent to which SOXB1 proteins can substitute genetically for one another in pluripotent cells in vivo.”

Discussion, fourth paragraph: This paper, which also shows co-expression and some functional redundancy of Sox2 and Sox3, should perhaps also be cited in relevant places: https://www.ncbi.nlm.nih.gov/pubmed/12514105

This paper by Avilion et al. is the seminal Sox2 deletion study. In this paper the authors discuss the Sox2 phenotype in relationship to expression of Sox2 and Sox3 and hypothesise that the failure of *Sox2*-null embryos early in development is due to the fact that no other SoxB1 gene is active as early. This is hypothesized to be a general point that SoxB1 phenotypes occur in cells where the particular SoxB1 gene mutated is the sole expressed member of the SoxB1 group. These are important ideas and we have now properly referred to them in our manuscript.